# Protease-mediated PRC1 dissociation promotes H2AK119ub remodeling during stress responses

Wei Cui [ID][1,10], Qingyang Li [ID][1,2,10], Jinsong Wei [ID][1,10], Ting Zhou[1,2,10], Haozhe Zhu[1], Delai Huang [ID][1,3], Shuai Wang[1], Jianan Gao[1], Ru Zhou[1], Zeyu Sun[4,5], Hua Ruan[6], Li Jan Lo [ID][1], Ting Tao[7,8], Jun Chen [ID][9], Jinrong Peng [ID][1] & Hui Shi [ID][1,2 ✉]

## Abstract

Chromatin must remain competent for responding rapidly to stress signals. Polycomb repressive complex 1 (PRC1) contributes to chromatin compaction through mono-ubiquitylation of histone H2A at Lys119 (H2AK119ub), yet how PRC1 complexes are dissociated from chromatin is only incompletely understood. Here, we show that the protease CAPN3 promotes rapid dissociation of PRC1 complexes from chromatin in response to stress. Following liver injury or heat shock, CAPN3 becomes activated and proteolyzes non-core PRC1 subunits to release the complexes from chromatin, leading to a reduction of H2AK119ub levels. These findings demonstrate that CAPN3 facilitates the remodeling of the chromatin landscape, unveiling a protease-mediated epigenetic mechanism for chromatin remodeling, and reveal CAPN3 as a key regulator of stress-induced epigenetic responses.

**Keywords** PRC1; H2AK119ub; CAPN3; Liver Regeneration; Stress
**Subject Categories** Chromatin, Transcription & Genomics; Post-translational Modifications & Proteolysis

## Introduction

Epigenetic regulation of gene expression has emerged as a key mechanism to control and maintain cell identity during multiple important biological processes including development, homeostasis, oncogenesis, and regeneration (Atlasi and Stunnenberg, 2017; Duan et al, 2022; Feinberg and Levchenko, 2023; Feinberg et al, 2006; Kumar et al, 2016; Liu et al, 2016; Sun et al, 2023; Wu et al, 2023; Xiang et al, 2020). The polycomb group (PcG) proteins form two major polycomb complexes with histone modification activities that are critical for gene silencing. Polycomb repressive complex 1 (PRC1) mono-ubiquitylates histone H2A at Lys119 (H2AK119ub) while Polycomb repressive complex 2 (PRC2) tri-methylates histone H3 at Lys27 (H3K27me3) (Blackledge et al, 2014; Cao et al, 2002; Czermin et al, 2002; de Napoles et al, 2004; Gao et al, 2012; Kuzmichev et al, 2002; Wang et al, 2004). In the last few decades, our understanding of how PRC1 and PRC2 complexes synergistically establish epigenetic landscapes has greatly expanded through a series of research works (Blackledge et al, 2014; Cooper et al, 2014; Cooper et al, 2016; Du et al, 2020; Endoh et al, 2017; Gao et al, 2012; Hauri et al, 2016; Kalashnikova et al, 2013; Kalb et al, 2014; Kasinath et al, 2021; Kassis and Brown, 2013; Li et al, 2017; McGinty et al, 2014; Mikkelsen et al, 2007; Pintacuda et al, 2017; Trojer et al, 2011; Wang et al, 2018; Zhao et al, 2020). However, how these complexes and their different variants are dissociated from chromatin have rarely been studied.

In mammals, PRC1 complexes are assembled around core components, either a RING1A or RING1B dimerized with one of the six PcG finger (PCGF1-6) subunits. Canonical PRC1 complexes (cPRC1) containing CBXs and PHCs only form around either PCGF2 or 4, while noncanonical PRC1 complexes (ncPRC1) containing RYBP/YAF2 form around all the PCGFs (Piunti and Shilatifard, 2021). cPRC1 are recruited by H3K27me3 to compact chromatin and repress gene expression, whereas ncPRC1 are recruited independently of PRC2 and H3K27me3, and function as the most active E3 ubiquitin ligases (Zhao et al, 2020; Lopez-Lacort et al, 2024). Different variants of PRC1 are recruited at different targets during various biological processes (Endoh et al, 2017; Gao et al, 2012; Hauri et al, 2016; He et al, 2013; Rose et al, 2016; Tardat et al, 2015; Wu et al, 2013).

The calpain system is one of the main intracellular proteolytic systems, primarily using proteolytic processing rather than degradation to modulate or modify substrates activity, specificity, longevity and localization (Storr et al, 2011). CAPN3, initially identified as a muscle-specific calpain, is among the sixteen calcium-activated cysteine proteases in the calpain family and is unique to vertebrates.

[1]MOE Key Laboratory of Biosystems Homeostasis & Protection, College of Animal Sciences, Zhejiang University, 310058 Hangzhou, China. [2]MOA Key Laboratory of Animal Virology, Zhejiang Provincial Engineering Research Center of Animal Biological Products, Zhejiang University Center for Veterinary Sciences, 310058 Hangzhou, China. [3]Department of Biology, University of Virginia, Charlottesville, VA 22904, USA. [4]Jinan Microecological Biomedicine Shandong Laboratory, Jinan 250117, China. [5]Yuhang Institute of Medical Science Innovation and Transformation, 310058 Hangzhou, China. [6]Key Laboratory of Freshwater Fish Reproduction and Development, Ministry of Education, State Key Laboratory Breeding Base of Eco-Environments and Bio-Resources of the Three Gorges Reservoir Region, School of Life Sciences, Southwest University, 400715 Chongqing, China. [7]Pediatric Cancer Research Center, National Clinical Research Center for Children and Adolescents' Health and Diseases, Children's Hospital Zhejiang University School of Medicine, 310052 Hangzhou, China. [8]Department of Surgical Oncology, Children's Hospital Zhejiang University School of Medicine, National Clinical Research Center for Children and Adolescents' Health and Diseases, 310052 Hangzhou, China. [9]College of Life Sciences, Zhejiang University, 310058 Hangzhou, China. [10]These authors contributed equally: Wei Cui, Qingyang Li, Jinsong Wei, Ting Zhou.✉E-mail: hui_shi@zju.edu.cn

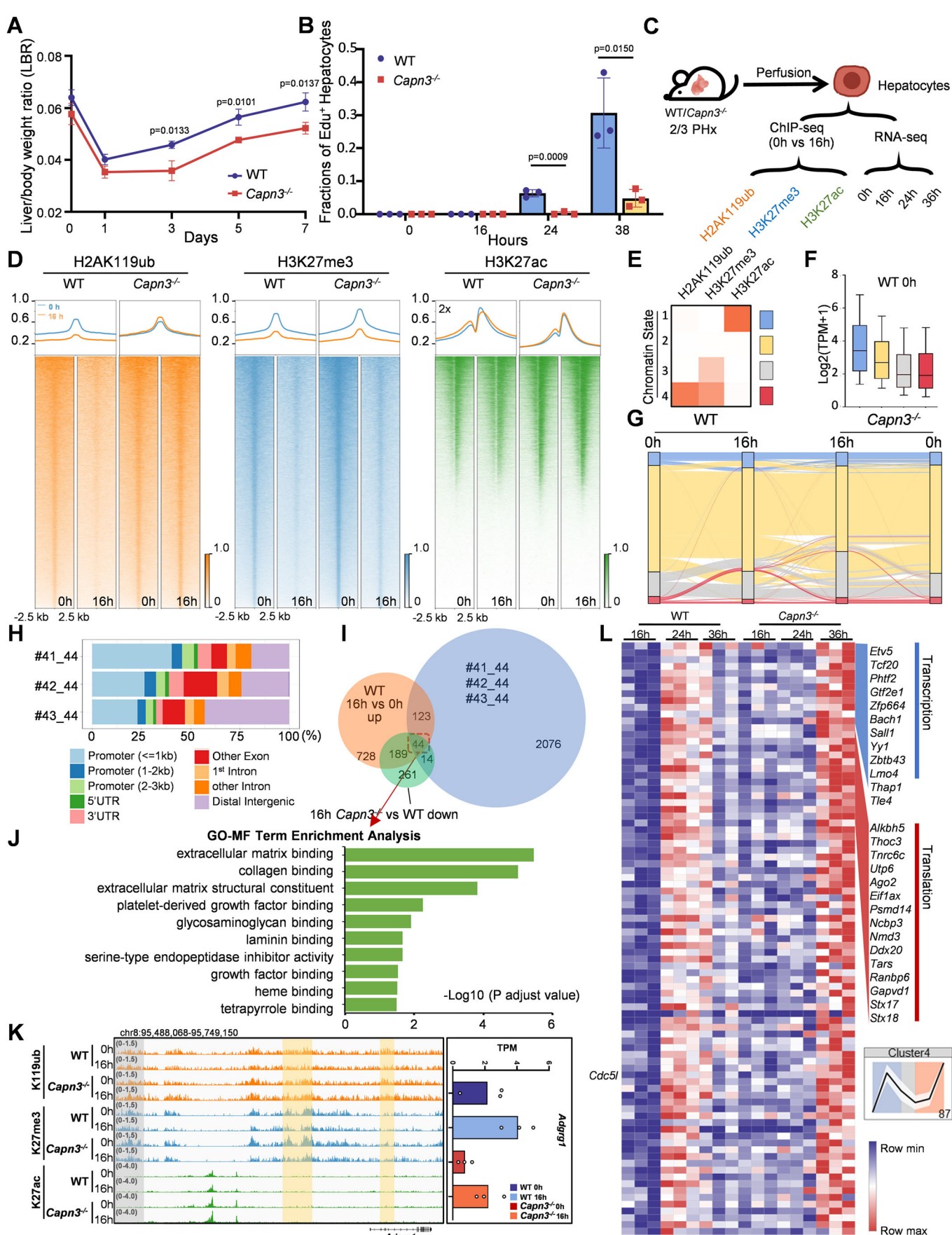

◄ **Figure 1. Depletion of CAPN3 inhibits H2AK119ub remodeling and delays the resumption of proliferation in hepatocytes after PHx.**

(A) The liver/body weight ratio (LBR) at different time points before and post-PHx in 2 months old wild-type (WT) and *Capn3*$^{-/-}$ mice. Data represent mean ± s.d. ($n = 3$). The statistical analysis was done using two-tailed Student's *t* test. A *P* value < 0.05 was considered statistically significant, and exact *P* values (*P*) are indicated in the figure. Source data are provided in the source data file. (B) Quantification of the fractions of EdU$^+$ hepatocytes at 0, 16, 24, and 38 h post-PHx. Data represent mean ± s.d. ($n = 3$). The statistical analysis was done using two-tailed Student's *t* test. A *P* value < 0.05 was considered statistically significant, and exact *P* values (*P*) are indicated in the figure. Source data are provided in the source data file. (C) Schematic showing experimental strategy. (D) ChIP-seq profiles and heatmaps of H2AK119ub, H3K27me3, and H3K27ac in WT and *Capn3*$^{-/-}$ hepatocytes at 0 h and 16 h post-PHx at called peaks (H2AK119ub and H3K27me3) or transcription start sites (TSS) (H3K27ac). Color intensity for each strand represents counts per million (CPM). (E) Heatmap illustrating four chromatin states categorized by ChromHMM based on three histone modifications (H2AK119ub, H3K27me3 and H3K27ac). The intensity of the orange color indicates the enrichment of each ChIP-seq signal belonging to the given chromatin state. (F) Box plot showing the expression levels (Transcripts per million, TPM) of genes within each chromatin state in WT hepatocytes at 0 h post-PHx. The center line indicates the median (50th percentile). The box bounds mark the first (Q1, 25th percentile) and third quartiles (Q3, 75th percentile). The whiskers extend to the most extreme data points within 1.5 times the interquartile range (IQR = Q3–Q1) from the quartiles. (G) Alluvial plots showing the global dynamics of chromatin states during liver regeneration (0 h vs 16 h post-PHx). Each line represents a 5 kb bin defined on the chromatin state categories. (H) Distribution of different subsets of regions overall genome obtained by ChIP-seq experiments. The four numerical codes for genomic regions denote the chromatin state classifications across the four conditions: WT-PHx-0 h, WT-PHx-16 h, *Capn3*$^{-/-}$-PHx-0 h, and *Capn3*$^{-/-}$-PHx-16 h. For instance, #41_44 represents "4 → 1 in WT vs.4 → 4 in *Capn3*$^{-/-}$", with the other codes following the same convention. (I) Venn diagrams showing the extent of overlap for genes near #41_44, #42_44 and #43_44 loci that display a delayed reduction in H2AK119ub, genes upregulated in WT hepatocytes at 16 h post-PHx vs. 0 h, and genes downregulated in *Capn3*$^{-/-}$ hepatocytes at 16 h post-PHx relative to WT. The intersection of these three gene sets is highlighted by the red dotted-line frame. (J) GO term enrichment analysis of the 44 genes in (I), *P* adjust values were determined by Benjamini–Hochberg correction. (K) Genome browser snapshots (left) of H2AK119ub, H3K27me3 and H3K27ac tracks at the *Adgrg1* loci in WT and *Capn3*$^{-/-}$ hepatocytes at 0 h or 16 h post-PHx. The expression levels (right) of *Adgrg1* in TPM at 0 h or 16 h post-PHx are summarized in bar charts. The yellow shaded areas highlight #42_44 regions, while the grey shaded area marks a #44_44 region with H2AK119ub signals unchanged across the four conditions. (L) Heatmap showing the 87 delayed upregulated genes in the *Capn3*$^{-/-}$ in Cluster4 defined by Mclust in Fig. EV2 (G). Source data are available online for this figure.

It functions as a calcium-activated protease and undergoes extremely rapid and exhaustive autolytic activity, which is abolished when its catalytic core residue is mutated (Ono et al, 2016; Sorimachi et al, 1989). Our previous studies in zebrafish indicated that CAPN3 may play an important role in liver development and regeneration, but the underlying mechanisms are not entirely clear (Chen et al, 2020; Ma et al, 2019; Tao et al, 2013).

In adult mammals, the liver is the only internal solid organ that can fully regenerate after resection. After 2/3 partial hepatectomy (PHx), the liver regenerates predominantly through hepatocyte proliferation (Michalopoulos, 2007). Recently, several studies have demonstrated the important roles of epigenetic landscape remodeling during liver regeneration, but most of the studies were primarily concentrated on the stages after the resumption of proliferation (Li et al, 2019; Zhang et al, 2021). It is reasonable to hypothesize that the chromatin of quiescent adult hepatocytes acquires competence in response to proliferative signals during liver regeneration.

In this study, we initially analyzed the global dynamics of chromatin modifications at 16 h post 2/3 PHx, a time point preceding the resumption of hepatocyte proliferation in mice (Michalopoulos, 2007). We found that depletion of CAPN3 inhibits the reduction of H2AK119ub and delays the resumption of hepatocyte proliferation after PHx. We further demonstrated that the proteolysis of PRC1 non-core components by CAPN3 facilitates this process. Additionally, we revealed that CAPN3 also promotes PRC1 dissociation from chromatin in response to heat shock, indicating a broader role of CAPN3 in chromatin remodeling under stress conditions.

## Results

### Depletion of CAPN3 inhibits H2AK119ub remodeling and further delays the resumption of proliferation in hepatocytes after PHx

To assess the role of CAPN3 during liver regeneration in mammals, a *Capn3*$^{-/-}$ mouse strain was constructed using a multiple sgRNAs-

directed CRISPR/Cas9 system to delete the exon2-22 region of *Capn3*. The genotypes were identified by PCR and confirmed by western blot analysis for the hepatocytes (Fig. EV1). We performed 2/3 PHx in both wild-type and *Capn3*$^{-/-}$ male mice and analyzed the liver-to-body weight ratios (LBR) at 0 (before), 1, 3, 5 and 7 days post-PHx (Fig. EV2A). Consistent with the loss of function of Capn3b in zebrafish (Chen et al, 2020), the *Capn3*$^{-/-}$ mice showed a clear delay in the recovery of LBR (Fig. 1A), indicating a critical role of CAPN3 in liver regeneration. We then utilized EdU pulse-labeling combined with Hnf4α immunofluorescence staining to quantify the proportion of hepatocytes re-entering the proliferative cycle. Compared to wild-type mice, the fractions of EdU-positive hepatocytes in *Capn3*$^{-/-}$ mice were significantly reduced at both 24 and 38 h (h) post-PHx (0.2% vs 6.9%, $P = 0.0009$ and 3.3% vs 30.8%, $P = 0.015$, respectively), while neither genotype exhibited detectable EdU-positive hepatocytes at 0 h and 16 h post-PHx (Figs. 1B and EV2B,C). This significant reduction in proliferating hepatocytes suggests that CAPN3 is essential for the timely resumption of hepatocyte proliferation during liver regeneration.

To determine whether chromatin states are repatterned prior to the resumption of hepatocyte proliferation in wild-type mice, and to assess whether this process is impaired in *Capn3*$^{-/-}$ mice, leading to delayed proliferation, hepatocytes from wild-type and *Capn3*$^{-/-}$ mice were isolated at 0 h and 16 h post-PHx, respectively. Subsequently, these cells were subjected to chromatin immunoprecipitation sequencing (ChIP-seq) for two repressive markers (H2AK119ub and H3K27me3) and an active marker (H3K27ac) (Fig. 1C). Both overall H2AK119ub and H3K27me3 ChIP-seq signals were remarkably decreased at 16 h post-PHx in wild-type hepatocytes, indicating the chromatin states were repatterned before the resumption of hepatocyte proliferation. For *Capn3*$^{-/-}$ hepatocytes, while the patterns of H3K27me3 and H3K27ac signals displayed patterns similar to that observed in wild-type hepatocytes, the overall H2AK119ub ChIP-seq signal was slightly accumulated rather decreased in *Capn3*$^{-/-}$ hepatocytes at 16 h post-PHx compared with 0 h. In other words, loss of function of

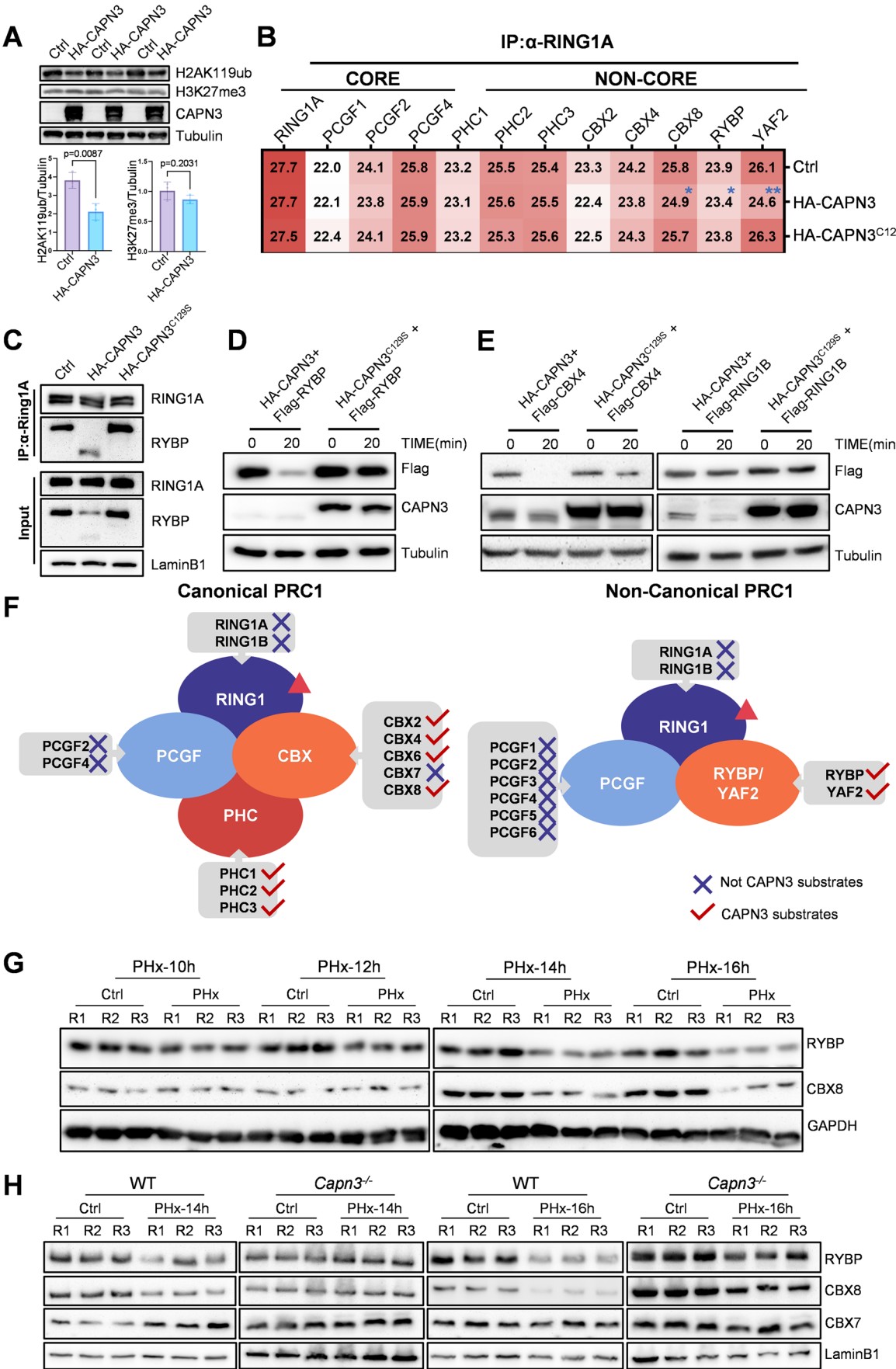

**Figure 2.   CAPN3 regulates H2AK119ub remodeling by proteolysis of PRC1 non-core components during liver regeneration.**

(A) Western blot analysis of three independent treatments of Ctrl and Tet-HA-CAPN3 HepG2 cells with 1 μg/mL doxycycline for 48 h (upper panel). Antibodies used are indicated. Lower panels show the quantification of relative expression levels of H2AK119ub (left) and H3K27me3 (right), normalized against Tubulin base on the western blot data. Data represent mean ± s.d. ($n = 3$). The statistical analysis was done using two-tailed Student's $t$ test. A $P$ value < 0.05 was considered statistically significant, and exact $P$ values ($P$) are indicated in the figure. Source data are provided in the source data file. (B) The Mass spectrometry (MS) analysis of RING1A immunoprecipitation (IP) from nuclear protein extracts of Ctrl, Tet-HA-CAPN3 and Tet-HA-CAPN3$^{C129S}$ HepG2 cells treated with 1 μg/mL doxycycline for 48 h. Each sample contains about $2 \times 10^7$ cells. The numbers and color intensity indicate the mean of log2-transformed Label-Free Quantification (LFQ) intensity values detected in the MS analysis. $n = 3$, *$P < 0.05$; **$P < 0.01$. The statistical analysis was done using two-tailed Student's $t$ test. Source data with exact $P$ values are provided in the source data file. (C) Western blot analysis of RING1A IP from nuclear protein extracts of Ctrl, Tet-HA-CAPN3 and Tet-HA-CAPN3$^{C129S}$ HepG2 cells treated with 1 μg/mL doxycycline for 48 h. Each sample contains about $2 \times 10^7$ cells. 'Input' contains ~10% of the input cell lysate used for IP. Antibodies used are indicated. This experiment was replicated at least twice. (D, E) Western blot analysis of in vitro CAPN3 proteolysis assays of RYBP, CBX4, and RING1B. The incubation times and antibodies used are indicated. These experiments were replicated at least twice. (F) Summary of in vitro CAPN3 proteolysis assays of PRC1 components. Most of PRC1 non-core components including PHC1/2/3, CBX2/4/6/8, RYBP and YAF2 (excepted CBX7) are CAPN3 substrates, while the core components including RING1A/B, PCGF1/2/3/4/5/6 are not. (G) Western blot analysis was performed on wild-type liver tissues collected at 10, 12, 14, 16 h post-PHx, using their excised tissues as control (Ctrl). Antibodies used are indicated. (H) Western blot analysis was performed on wild-type and *Capn3*$^{-/-}$ liver tissues collected at 14 and 16 h post-PHx, using their excised tissues as control. Antibodies used are indicated. Source data are available online for this figure.

CAPN3 specifically inhibited overall H2AK119ub reduction in hepatocytes at 16 h post-PHx (Fig. 1D). The differential dynamics of H2AK119ub between wild-type and *Capn3*$^{-/-}$ hepatocytes at 16 h post-PHx were further confirmed by CUT&Tag analysis and western blot analysis of chromatin fractions (Fig. EV2D,E).

We next utilized ChromHMM to classify genomic patterns of wild-type and *Capn3*$^{-/-}$ hepatocytes at 0 h and 16 h post-PHx into four chromatin states according to the three histone modifications (Fig. 1E; Dataset EV1). Chromatin state 1, characterized by H3K27ac alone, corresponded to higher gene expression levels. In contrast, chromatin state 3 and 4, marked by repressive modifications, corresponded to lower gene expression levels (Fig. 1F). The alluvial diagram was used to illustrate the global dynamics of chromatin states preceding the resumption of hepatocyte proliferation during liver regeneration (0 h vs 16 h post-PHx). Each line in the diagram represents a 5 kb chromatin region, tracing the chromatin states of 0 h wild-type, 16 h wild-type, 16 h *Capn3*$^{-/-}$ and 0 h *Capn3*$^{-/-}$. The analysis revealed that the chromatin states of some regions remained unchanged in *Capn3*$^{-/-}$ hepatocytes while the same regions were remodeled in wild-type hepatocytes during this period. In particular, a subset of regions transitioned from state 4 to state 1, 2, or 3 in wild-type hepatocytes at 16 h post-PHx, indicating that the H2AK119ub and H3K27me3 signals were greatly decreased at this time point. However, these regions stayed classified as state 4 in both 0 h and 16 h post-PHx in *Capn3*$^{-/-}$ hepatocytes (called #41_44, #42_44, and #43_44), indicating that loss of function of CAPN3 hindered the removal of H2AK119ub and consequently influenced the remodeling of other histone modifications at these regions (Fig. 1G). The predominant fractions within this subset of loci are categorized as promoter and distal intergenic regions, suggesting their potential roles in the regulatory landscape (Fig. 1H). To evaluate whether such disruption of chromatin remodeling would further affect downstream gene expression at 16 h or later, we performed RNA-seq analysis on wild-type and *Capn3*$^{-/-}$ hepatocytes at 0, 16, 24, and 36 h post-PHx (Fig. 1C). By integrating the ChIP-seq and transcriptomic data at 0 h and 16 h, we identified 44 genes proximal to these disrupted loci whose induction at 16 h was significantly attenuated in *Capn3*$^{-/-}$ hepatocytes compared to wild-type hepatocytes, suggesting their full activation may relate to the chromatin remodeling (Figs. 1I and EV2F). Gene Ontology (GO) analysis of the 44 genes revealed significant enrichment for molecular functions (MF) related to

extracellular matrix interactions (e.g. collagen, glycosaminoglycan, and laminin binding, extracellular matrix structural constituent), growth factor signaling modulation (platelet-derived growth factor and growth factor binding), and key metabolic and regulatory functions (heme and tetrapyrrole binding, serine-type endopeptidase inhibitor activity), suggesting a role in preparing for signal responses (Fig. 1J,K).

To further assess the later impact of chromatin remodeling disruption at 16 h post-PHx on downstream genes, we subjected genes annotated to #41_44, #42_44, and #43_44 (detected at >0.1 TPM in at least one conditions of the 16, 24, and 36 h) to model-based clustering using Mclust into 16 clusters (Fig. EV2G,H) (Scrucca et al, 2016). Cluster 7 and 16 showed increased gene expression at 24 h and 36 h in wild-type hepatocytes, while gene expression levels remained stable across 16, 24, and 36 h in *Capn3*$^{-/}$$^{-}$ hepatocytes (Fig. EV2H). Interestingly, Cluster4 exhibited delayed expression in *Capn3*$^{-/-}$ hepatocytes, peaking at 36 h instead of 24 h as seen in wild-type hepatocytes, including transcription related genes (e.g. *Etv5*, *Tcf20*, *Phtf2*), mRNA maturation related genes (e.g. *Alkbh5*, *Thoc3*, *Tnrc6c*), protein synthesis and transport related genes (e.g. *Utp6*, *Ago2*, *Tars*), and a crucial cell cycle regulator, *Cdc5l* (Figs. EV2H and 1L). These findings suggest that the disruption of chromatin remodeling in *Capn3*$^{-/-}$ hepatocytes at 16 h post-PHx continues to impair the subsequent induction of downstream genes during later stages of liver regeneration.

## CAPN3 regulates H2AK119ub remodeling by proteolysis of PRC1 non-core components during liver regeneration

To assess the role of CAPN3 in H2AK119ub deubiquitylation, we generated a doxycycline-inducible HA-CAPN3 expression HepG2 cell line (Tet-HA-CAPN3). After 2 days of doxycycline treatment, H2AK119ub levels decreased significantly, whereas H3K27me3 levels remained unchanged, suggesting that CAPN3 specifically promotes H2AK119ub deubiquitylation (Fig. 2A). Given that H2AK119ub is modified by PRC1 complexes, we hypothesized that CAPN3 activation might affect the integrity or assembly of PRC1 complexes. To further verify our hypothesis, we generated another HepG2 cell line with doxycycline-inducible HA-CAPN3$^{C129S}$ expression (Tet-HA-CAPN3$^{C129S}$). CAPN3$^{C129S}$ is an enzymatic inactive mutant of CAPN3 whose catalytic core residue Cys$^{129}$ was mutated to serine, serving as a negative control

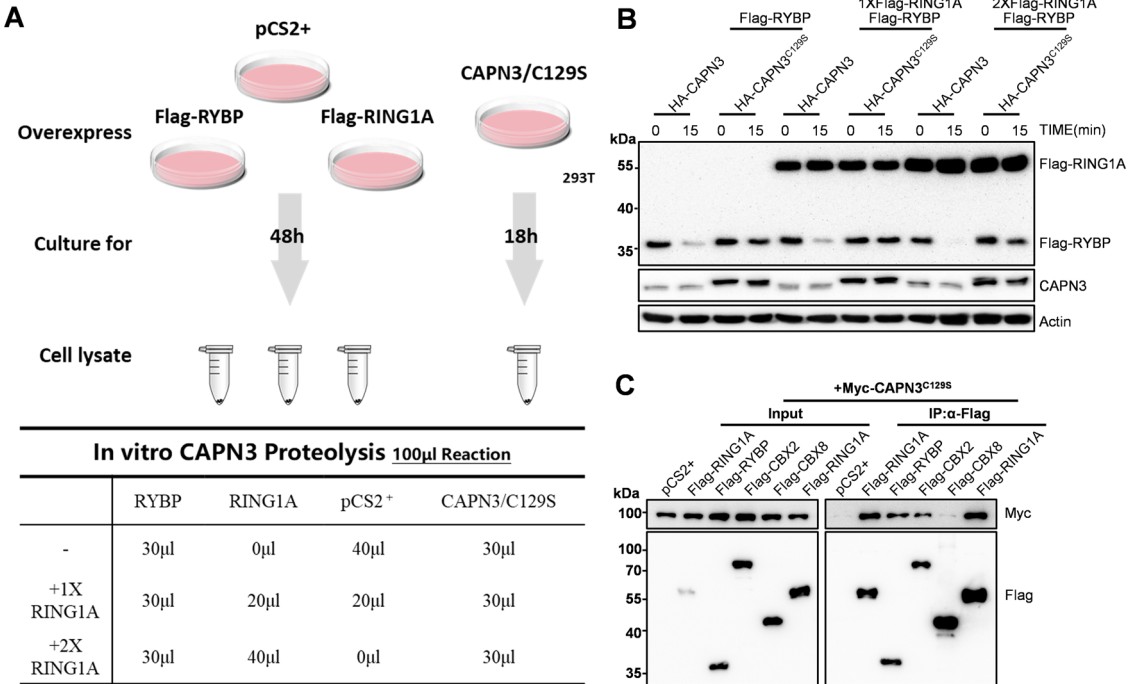

**Figure 3. PRC1 core component RING1A helps to recruit CAPN3 to proteolyze non-core components.**

(A) Schematic showing experimental strategy. (B) Western blot analysis of in vitro CAPN3 proteolysis assays of RYBP with different concentration of RING1A. The reaction of each sample is shown in the table of (A). Antibodies used are indicated. This experiment was replicated at least twice. (C) Western blot analysis on anti-Flag immunoprecipitated from cell lysates using anti-Myc and anti-Flag antibodies. 293 T cells were co-transfected with 500 ng Myc-CAPN3^C129S-pCS2+ and 1ug pCS2+, 500 ng Flag-RING1A-pCS2+, 500 ng Flag-RYBP-pCS2+, 500 ng Flag-CBX2-pCS2+, 500 ng Flag-CBX8-pCS2+ or 1ug Flag-RING1A-pCS2+. Cell lysates were harvested after 48 h. 'Input' contains ~10% of the input cell lysate used for IP. This experiment was replicated at least twice. Source data are available online for this figure.

(Sorimachi et al, 1993; Tao et al, 2013). After 48 h of doxycycline treatment, Ctrl, Tet-HA-CAPN3 and Tet-HA-CAPN3^C129S cells were harvested, and the nuclear protein fraction was subsequently extracted. The nuclear extracts were affinity purified by RING1A antibody and then analyzed by mass spectrometry (MS). The MS analysis revealed that overexpression of CAPN3 resulted in a significant reduction in several non-core PRC1 components, including CBX8, RYBP and YAF2 (Fig. 2B). Further western blot analysis confirmed that RYBP protein levels decreased in both nuclear extracts and RING1A-PRC1 complexes upon CAPN3 overexpression. Notably, a shorter form of RYBP was detected, suggesting that RYBP might be a substrate of CAPN3 (Fig. 2C). The in vitro CAPN3 proteolysis analysis showed that CAPN3 cleaved RYBP efficiently (Fig. 2D). To identify other potential substrates of CAPN3 within PRC1 complexes, we cloned all known main PRC1 components (total 18, including family members of the core components RING1 and PCGF and the family members of non-core components CBX, PHC and RYBP). The in vitro CAPN3 proteolysis assays revealed that all tested non-core PRC1 components (except CBX7) were obviously cleaved by CAPN3, while all the core PRC1 components were not (Figs. 2E,F and EV3).

To validate this mechanism in liver regeneration, we collected wild-type liver samples every two hours from 10 h to 16 h post-PHx. Western blot analysis showed that PRC1 components such as RYBP and CBX8 were obviously decreased at 14 h and 16 h post-PHx (Fig. 2G). In contrast, this decrease was inhibited in Capn3^−/−

hepatocytes at both 14 h and 16 h post-PHx, confirming the regulatory role of CAPN3 in mediating PRC1 components degradation during liver regeneration (Fig. 2H).

## PRC1 core component RING1A helps to recruit CAPN3 to proteolyze non-core components

As shown in Fig. 2C, both the reduction of full-length RYBP and the ratio of truncated RYBP within PRC1 complexes was greater than that in the total protein content upon CAPN3 overexpression. We suspect that the assembled of PRC1 complexes might facilitate the proteolysis of PRC1 non-core components by CAPN3 (Fig. 2C). To verify this hypothesis, we performed in vitro CAPN3 proteolysis assays of Flag-RYBP with varying concentration of Flag-RING1A (0, 1× and 2×). We found that the residual Flag-RYBP was inversely related to the amount of Flag-RING1A present in the reaction 15 min after mixing with CAPN3. Thus, the PRC1 core component RING1A appears to promote the proteolysis of PRC1 non-core components (at least RYBP) by CAPN3 (Fig. 3A,B). To test whether this effect is due to a higher affinity of RING1A for CAPN3, facilitating its recruitment, we overexpressed Myc-tagged CAPN3^C129S together with Flag-tagged RING1A and several PRC1 non-core components and performed co-immunoprecipitation analyses. The results showed that CAPN3^C129S had a higher binding affinity for the PRC1 core component RING1A compared to non-core components like RYBP, CBX2 and CBX8. This suggests that the PRC1 core component RING1A may facilitate the recruitment

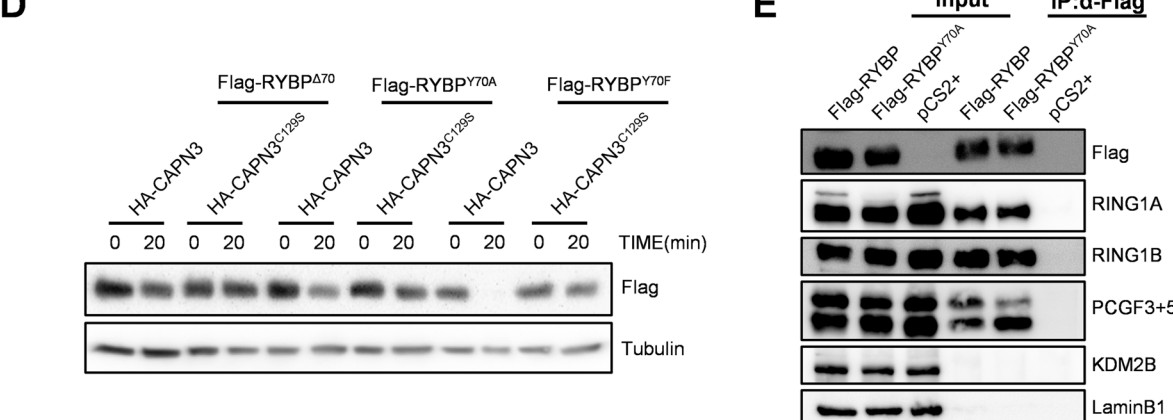

**A**

NZF C-terminal binding motif

**B**

**C**

**D**

**E**

**Figure 4.  Tyr[70] of RYBP was the key amino acid for its proteolysis by CAPN3.**

(A) Structural diagram of RYBP mutant with different amino acid deletion. Mutants that cannot be proteolytic are highlighted in orange. hRD1 and hRD2 deleted the Npl4 zinc finger (NZF) domain and C-terminal binding motif of RYBP, respectively. (B) Western blot analysis of in vitro CAPN3 proteolysis assays of 6 different single amino acid deletion mutants of RYBP. The incubation times and antibodies used are indicated. This experiment was replicated at least twice. (C) RYBP protein sequence alignment and phylogenetic analysis. The conserved Tyrosine residues are highlighted in red. Species with CAPN3 proteins are highlighted in red. (D) Western blot analysis of in vitro CAPN3 proteolysis assays of point mutations of RYBP. The incubation times and antibodies used are indicated. This experiment was replicated at least twice. (E) Western blot analysis on anti-Flag immunoprecipitated from nuclear extracts of 293 T cells transfected with 500 ng *Flag-RYBP-pCS2+*, 500 ng *Flag-RYBP[Y70A]-pCS2+* or 500 ng *pCS2+*. Cell lysates were harvested after 48 h. 'Input' contains ~10% of the input cell lysate used for IP. Antibodies used are indicated. Source data are available online for this figure.

of CAPN3 for the proteolysis of PRC1 non-core components (Fig. 3C).

## RYBP[Y70A] mutant protein is insensitive to CAPN3 proteolytic activity

Our data showed that depletion of CAPN3 inhibits H2AK119ub remodeling and further delays proliferation in hepatocytes after PHx (Fig. 1). As RYBP-PRC1 recruitment is independent of PRC2 and exhibits strong catalytic activity in mediating H2AK119ub propagation (Bernstein et al, 2006; Cooper et al, 2014; Gao et al, 2012; He et al, 2013; Rose et al, 2016; Simon and Kingston, 2013; Tavares et al, 2012; Wu et al, 2013), we aimed to determine if CAPN3-mediated proteolysis of RYBP is critical for timely liver regeneration. Previous studies reported a primary sequence motif underlying CAPN3 substrates for proteolysis: [LIMV]1-X2-X3-X4-X5-[LIMV]6-X7-X8-[LIMV]9-[DE]10 (de Morrée et al, 2010). However, RYBP lacks these classic CAPN3 recognition sites. To identify the crucial regions for CAPN3 proteolysis, we firstly generated a series of *RYBP* deletion constructs (*hRD1-hRD11*) and performed the in vitro CAPN3 proteolysis assay. The results pinpointed the CAPN3 proteolysis site between the 67th and 75th amino acids of RYBP (Figs. 4A and EV4). Further fine mapping revealed that deletion of a single amino acid at the 70th position (Tyr[70]) in RYBP resulted in the RYBP-ΔTyr[70] mutant protein to become insensitive to CAPN3-mediated proteolysis (Fig. 4B). Interestingly, by protein sequence alignment and phylogenetic analysis, we found that this Tyr[70] site of RYBP is conserved only in vertebrates, coinciding with the emerging of the *CAPN3* gene in vertebrates (Fig. 4C) (Ono and Sorimachi, 2015). Additionally, substituting Tyr[70] with alanine (Y70A), but not with phenylalanine (Y70F) rendered RYBP resistant to CAPN3 proteolysis (Fig. 4D). To further assess whether the RYBP[Y70A] mutation preserves the properties of PRC1 components, we overexpressed either Flag-RYBP or Flag-RYBP[Y70A] in 293 T cells and performed immunoprecipitation using anti-Flag beads. Subsequent western blot analysis revealed that both wild-type and mutant RYBP co-immunoprecipitated with RING1A/B and PCGF3/5, indicating that RYBP[Y70A] maintains its ability to form PRC1 complexes (Fig. 4E).

## RYBP[Y70A] knock-in mice exhibit delayed liver regeneration after PHx

We further generated a knock-in mouse strain carrying the RYBP[Y70A] mutation (Fig. 5A). Western blot analysis revealed that homozygous RYBP[Y70A] mutant livers exhibited slightly elevated

RYBP expression levels and comparable H2AK119ub levels relative to wild-type livers (Fig. 5B). To evaluate whether liver regeneration was affected in *RYBP[Y70A]* knock-in mice, we conducted 2/3 PHx on both wild-type and homozygous *RYBP[Y70A]* knock-in mice. LBR were measured at 0, 1, 3 and 5 days post-PHx. The results showed a significant delay in LBR recovery during liver regeneration in homozygous *RYBP[Y70A]* knock-in mice (Fig. 5C). Consistent with the results from *Capn3[−/−]* mice, *RYBP[Y70A]* homozygous mice exhibited a significant reduction in hepatocyte proliferation, as indicated by EdU pulse-labeling, at 72 h post-PHx compared to wild-type mice (27.95% in wild-type vs. 5.58% in homozygous *RYBP[Y70A]* knock-in mice, *P* < 0.001), further supporting the role of RYBP proteolysis by CAPN3 in the resumption of hepatocyte proliferation after PHx (Fig. 5D,E).

## Loss of function of CAPN3 delays the dissociation of PRC1 from chromatin during stress response

It was previously shown that upon heat shock, various PRC1 components were transferred into the nucleolus (Azkanaz et al, 2019). This relocation coincided with a dramatic loss of H2AK119ub but not H3K27me3, which similar to the outcomes of our CAPN3 overexpression experiments in this study (Fig. 2A). Here, we isolated primary hepatocytes from wild-type mice and cultured them for 24 h. Subsequently, the primary hepatocytes were heat shocked by incubation at 45 °C for 60 min (min). After heat shock treatment, only the levels of H2AK119ub exhibited a noticeable decrease, whereas the levels of H3K27me3 and H3K27ac remained unchanged (Fig. 6A).

To assess whether CAPN3 plays a role in this process, we incubated both wild-type and *Capn3[−/−]* primary hepatocytes at 45 °C for 0, 15, 30, or 60 min for further analysis. Immunofluorescence analysis revealed a significant delay in the transfer rate of RYBP to the nucleolus in *Capn3[−/−]* primary hepatocytes (Figs. 6B and EV5A–D). We hypothesized that CAPN3 promotes PRC1 dissociation from chromatin in response to stress signals such as PHx and heat shock, thus delaying the nucleolar translocation of RYBP. To validate this hypothesis, we isolated chromatin fractions for subsequent western blot analysis. In wild-type primary hepatocytes, the levels of chromatin-bound PRC1 non-core components rapidly decreased upon heat shock, whereas this process was markedly delayed in *Capn3[−/−]* cells (Fig. 6C). Moreover, the levels of chromatin-bound PRC1 core component RING1A also rapidly declined after heat shock in wild-type cells, while the abundance of RING1A in the total cell extracts and nuclear extracts remained unchanged (Fig. 6D). The reduction of chromatin-bound RING1A and H2AK119ub were both inhibited in

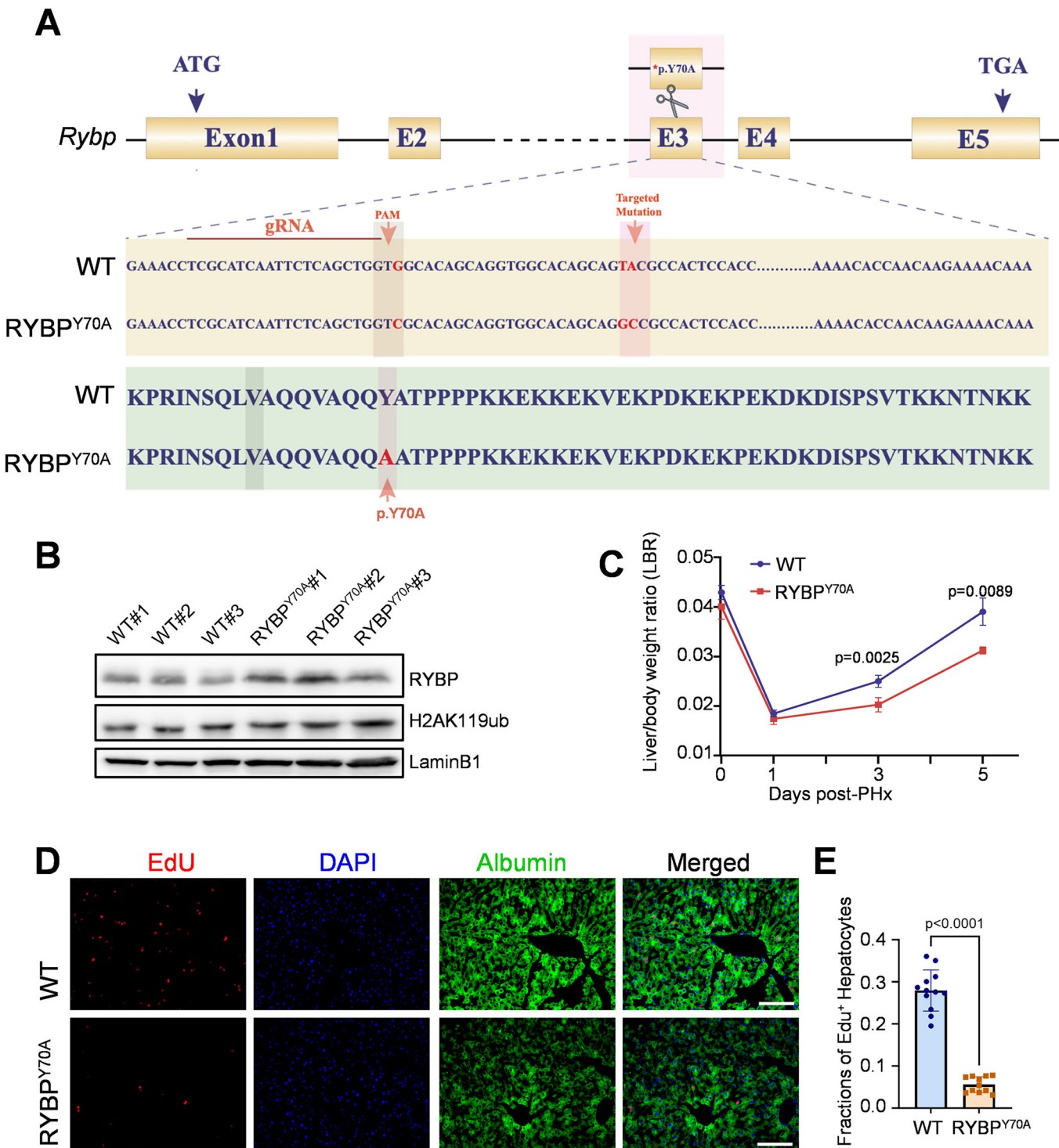

**Figure 5.** *RYBP^{Y70A}* knock-in mice exhibit delayed liver regeneration after PHx.

(A) Schematic showing the strategy of the construction of *RYBP^{Y70A}* knock-in mice and the alignments of nucleotide sequences and protein sequence (bottom) from wild-type (WT) and homozygous *RYBP^{Y70A}* knock-in mice. (B) Western blot analysis of wild-type and RYBP^{Y70A} knock-in mice livers. Antibodies used are indicated. (C) The liver/body weight ratio (LBR) at different time points before and post-PHx in 2-month-old WT and RYBP^{Y70A} mice. Data represent mean ± s.d. (n = 3–4). The statistical analysis was done using two-tailed Student's *t* test. A P value < 0.05 was considered statistically significant, and exact P values (P) are indicated in the figure. Source data are provided in the source data file. (D) Immunostaining of EdU and Albumin (a hepatocyte marker) at 72 h post-PHx of WT and RYBP^{Y70A} knock-in mice livers. Scale bars, 100 μm. (E) Quantification of the fractions of EdU⁺ hepatocytes at 72 h post-PHx. Data represent mean ± s.d. (n = 11–12 random fields from two mice per genotype). The statistical analysis was done using two-tailed Student's *t* test. A P value < 0.05 was considered statistically significant, and exact P values (P) are indicated in the figure. Source data are provided in the source data file. Source data are available online for this figure.

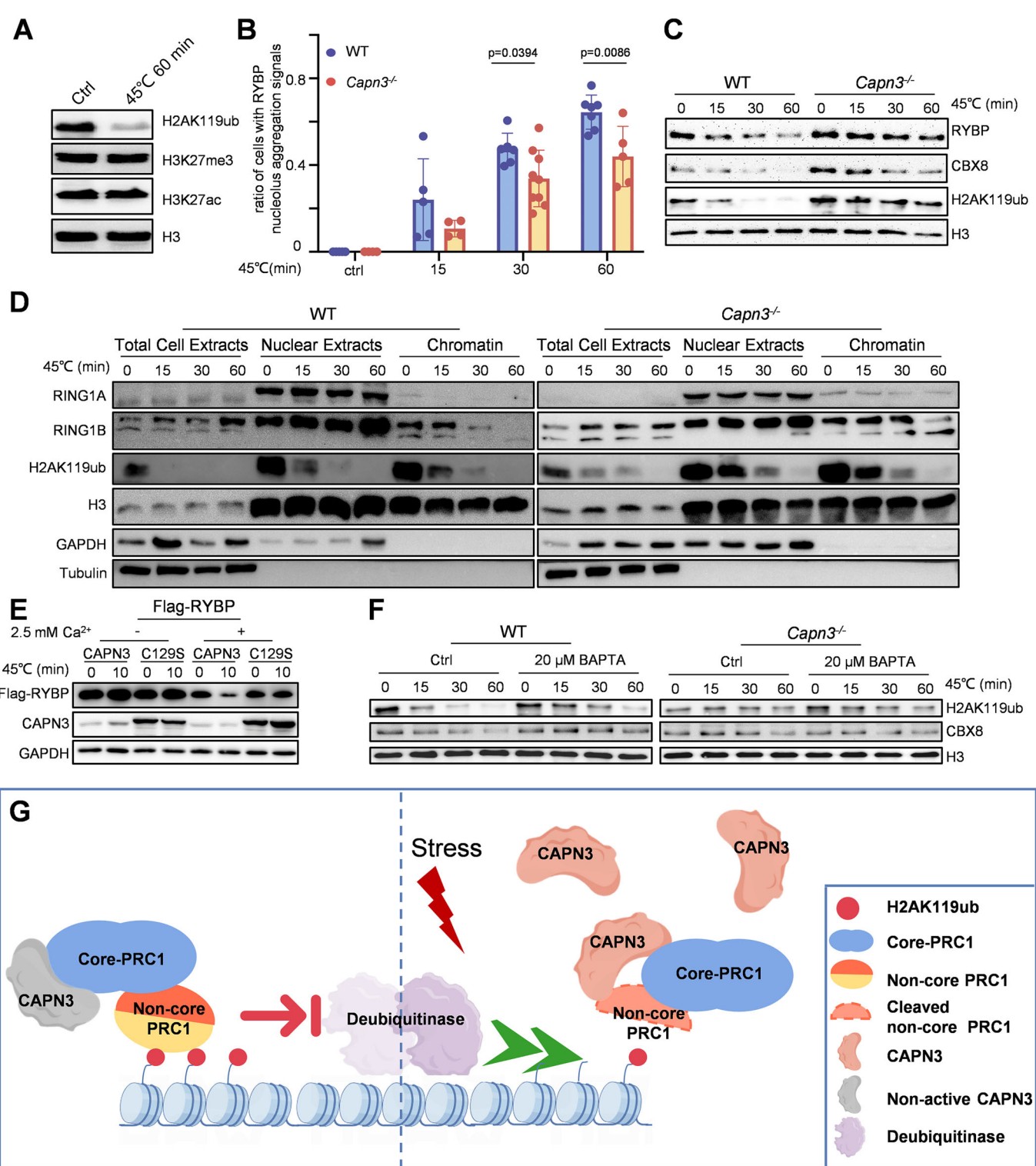

$Capn3^{-/-}$ cells, indicating that CAPN3 promotes the dissociation of PRC1 from chromatin and the subsequent reduction of H2AK119ub following heat shock (Fig. 6C,D). A similar delayed PRC1 dissociation from chromatin and H2AK119ub reduction after heat shock was observed in primary hepatocytes derived from homozygous $RYBP^{Y70A}$ knock-in mice, further supporting the role

of the CAPN3-PRC1 pathway in H2AK119ub reduction during stress response (Fig. EV5E).

We then sought to distinguish whether stress signals trigger the recruitment of CAPN3 to the PRC1 complex or activation of CAPN3 in the PRC1 complexes to compromise the chromatin-bound PRC1. Due to the ineffectiveness of the RING1A antibody

**Figure 6.   Proteolysis of PRC1 non-core components by CAPN3 disrupts the maintenance of genomic H2AK119ub by PRC1 occupation.**

(A) Western blot analysis of wild-type mice primary hepatocyte cells incubated at 45 °C for 0 or 60 min were performed with indicated antibodies. This experiment was replicated at least twice. (B) Quantification of fractions of RYBP nucleolar accumulations by immunofluorescence after heat shock (45 °C for 0, 15, 30, or 60 min) of primary hepatocytes isolated from WT and *Capn3*$^{-/-}$ mice. Data represent mean ± s.d. ($n$ = 4–9). The statistical analysis was done using two-tailed Student's $t$ test. A $P$ value < 0.05 was considered statistically significant, and exact $P$ values ($P$) are indicated in the figure. The source data are provided as a source data file. (C) Chromatin fractions were isolated from WT and *Capn3*$^{-/-}$ primary hepatocytes incubated at 45 °C for 0, 15, 30, or 60 min. Western blot analysis was performed with indicated antibodies. This experiment was replicated at least twice. (D) Total cell extracts, nuclear extracts and chromatin fractions were isolated from WT and *Capn3*$^{-/-}$ primary hepatocytes incubated at 45 °C for 0, 15, 30, or 60 min. Western blot analysis was performed with indicated antibodies. This experiment was replicated at least twice. (E) Western blot analysis of in vitro CAPN3 proteolysis assays of human RYBP. Reaction mixture (containing 2 mM EDTA, chelated endogenous $Ca^{2+}$) was incubated at 37 °C for 10 min with or without 2.5 mM $Ca^{2+}$ as indicated. This experiment was replicated at least twice. (F) Chromatin fractions were isolated from WT and *Capn3*$^{-/-}$ primary hepatocytes with/without 20 µM BAPTA pre-treated for 1 h and incubated at 45 °C for 0, 15, 30, or 60 min. Western blot analysis was performed with indicated antibodies. This experiment was replicated at least twice. (G) A model illustrates that PRC1 occupation of chromatin maintains H2AK119ub from deubiquitylation. CAPN3 becomes activated after stress and its proteolysis of PRC1 non-core components facilitates the release of PRC1 complexes from chromatin, resulting in a decrease in H2AK119ub levels. Source data are available online for this figure.

for immunoprecipitation in mice, we had to conduct an immunoprecipitation using a rat liver that was not subjected to any stress treatment, including PHx or heat shock. This assay revealed that the endogenous CAPN3 is capable of binding with PRC1 complexes prior to the imposing of stress signals (Fig. EV5F). CAPN3 is a calcium-dependent cysteine protease. We then performed the in vitro cleavage assay for RYBP in both the presence and absence of free calcium ions. The results indicated that RYBP was cleaved by CAPN3 only in the presence of free calcium ions, demonstrating that this reaction is calcium ion-dependent (Fig. 6E). Thus, we hypothesized that CAPN3 binds to PRC1 complexes prior to a response to stress signals and is likely activated by free calcium ions induced by cellular stresses. To test this hypothesis, we pre-treated hepatocytes with the calcium chelator BAPTA and incubated them at 45 °C for 0, 15, 30, or 60 min. Chromatin was extracted and western blot analysis was performed. The results indicated that BAPTA treatment delayed the elimination of both H2AK119ub and CBX8 from chromatin during heat shock, but not in *Capn3*$^{-/-}$ samples, suggesting that stress induced calcium fluctuation might play a key role in activating the CAPN3-PRC1 regulatory pathway (Fig. 6F).

Therefore, we proposed a model of the CAPN3-mediated PRC1 regulation system during H2AK119ub remodeling. The Polycomb Repressive-Deubiquitinase (PR-DUB) complex plays a pivotal role in removing ubiquitin from H2AK119ub on nucleosomes and counteracts the ubiquitin E3 ligase function of PRC1 (Ge et al, 2023). On the other hand, the PRC1 complexes maintains H2AK119ub from deubiquitylation. After stress events such as PHx and heat shock, CAPN3 becomes activated and facilitates the dissociation of PRC1 complexes from chromatin through its proteolytic activity on non-core PRC1 components. Once a H2AK119ub modified region loses PRC1 occupancy, it becomes susceptible to rapidly deubiquitylation (Fig. 6G).

## Discussion

The liver regenerates predominantly through hepatocyte proliferation after 2/3 PHx (Michalopoulos, 2007). Although numerous studies have shown that the epigenetic landscape plays an important role in liver regeneration, whether and how chromatin becomes competent to respond to stress signals has been rarely studied. Here, we employed a *Capn3*$^{-/-}$ mouse model, which exhibited a significant delay in both resumption of hepatocyte proliferation and recovery of LBR during liver regeneration, to investigate this issue.

We demonstrated that the global chromatin state landscape of hepatocyte needs to be repatterned before restarting proliferation. Despite the stability of overall H3K27ac signals, there was a remarkable reduction in overall levels of H2AK119ub and H3K27me3 at 16 h post-PHx, a time point preceding hepatocyte proliferation in mice. This reduction suggests a crucial role in priming the chromatin landscape for resumption of proliferation. By employing ChromHMM analysis, we characterized the chromatin into four states based on the three histone modifications. We then utilized an alluvial diagram to track the global dynamics of chromatin states, revealing the chromatin states repatterning during this biological process. However, in *Capn3*$^{-/-}$ hepatocytes, the chromatin repatterning was disrupted, particularly with inhibited reduction of overall H2AK119ub levels, which was concomitant with a delayed upregulation of a series of genes involved in the preparation for subsequent liver regeneration process.

We further demonstrated that most of the PRC1 non-core components, but not the core components, are CAPN3 substrates, and we observed a reduction in the protein levels of these components during liver regeneration. We focused on RYBP, a key component in the PRC1 complexes for achieving H2AK119ub, and identified the Tyr$^{70}$ amino acid to be indispensable for CAPN3-mediated cleavage. To verify the functional significance of this proteolytic regulation, we constructed a knock-in mouse model (RYBP$^{Y70A}$) with a single amino acid mutation that protects against the proteolysis by CAPN3. Notably, these mice exhibited similar defects in hepatocyte proliferation resumption and LBR recovery during liver regeneration as observed in *Capn3*$^{-/-}$ mice, indicating that the proteolysis of RYBP by CAPN3 is crucial for timely liver regeneration following PHx.

We then hypothesized that the CAPN3-mediated PRC1 dissociation is responsible for rapidly rendering chromatin competent to stress signals. Previous studies have indicated a prompt reduction in H2AK119ub levels, but not H3K27me3 levels, upon the translocation of PRC1 complexes to nucleoli (Azkanaz et al, 2019), which coincides with the outcome of CAPN3 overexpression in this study. Using primary hepatocytes, we revealed that CAPN3 promotes the dissociation of PRC1 from

chromatin and subsequent reduction of H2AK119ub after heat shock, further reinforcing and broadening the role of CAPN3 in chromatin remodeling under stress conditions.

In addition, we found that chelating calcium attenuated the effect of heat shock on reducing the H2AK119ub level. Considering that PHx can also elevate $Ca^{2+}$ signaling (Oliva-Vilarnau et al, 2018) and that CAPN3 complexes with the PRC1, we propose that calcium acts as an activator of the CAPN3-PRC1 pathway in response to stress signals such as PHx and heat shock. Overall, our study demonstrated the role of CAPN3 in facilitating the dissociation of PRC1 from chromatin during H2AK119ub remodeling, unveiling a novel protease-mediated epigenetic regulation system in rapid response to stress.

CAPN3 is a vertebrate specific gene, and we found that its proteolytic site on RYBP is also specifically conserved only in vertebrates. Although PRC1 complexes are crucial for gene silencing and chromatin remodeling in both vertebrates and invertebrates, the composition of PRC1 complexes in vertebrates is much more diverse and regulated by more complex developmental and environmental stimuli. CAPN3, as we showed here, plays a crucial role in facilitating this regulation.

However, as we observed in both PHx and heat shock treatment, loss of CAPN3 only delays but not completely blocks the reduction of chromatin-bound PRC1, suggesting that additional pathways are involved in this process. Nonetheless, as we have consistently emphasized, CAPN3 is critical for the rapid response to stress signals.

# Methods

### Reagents and tools table

| Reagent/resource | Reference or source | Identifier or catalog number |
| --- | --- | --- |
| **Experimental models** | | |
| Mouse: WT: C57BL/6JGpt-WT | Gem Pharmatech | N/A |
| Mouse: Capn3$^{-/-}$: C57BL/7JGpt-Capn3$^{-/-}$ | Gem Pharmatech | N/A |
| Mouse: RYBP$^{Y70A/Y70A}$: C57BL/7JGpt-RYBP$^{Y70A}$ | Gem Pharmatech | N/A |
| Rat: Sprague-Dawley | Gem Pharmatech | N/A |
| HEK293T cell | Cell Bank of the Chinese Academy of Sciences | N/A |
| HepG2 cell | Cell Bank of the Chinese Academy of Sciences | N/A |
| Tet-HA-CAPN3 HepG2 Cell | This study | N/A |
| Tet-HA-CAPN3$^{C129S}$ HepG2 Cell | This study | N/A |
| **Recombinant DNA** | | |
| pRetroX-TetOne-Puro | Clontech | Cat #634309 |
| pCS2 + -HA-CAPN3 | Zhao et al, 2019 | |
| pCS2 + -HA-CAPN3 $^{C129S}$ | Zhao et al, 2019 | |
| pCS2 + -Flag-RYBP | This study | |
| pCS2 + -Flag-CBX2 | This study | |
| pCS2 + -Flag-CBX4 | This study | |
| pCS2 + -Flag-CBX6 | This study | |

| Reagent/resource | Reference or source | Identifier or catalog number |
| --- | --- | --- |
| pCS2 + -Flag-CBX7 | This study | |
| pCS2 + -Flag-CBX8 | This study | |
| pCS2 + -Flag-YAF2 | This study | |
| pCS2 + -Flag-PHC1 | This study | |
| pCS2 + -Flag-PHC2 | This study | |
| pCS2 + -Flag-PHC3 | This study | |
| pCS2 + -Flag-RING1A | This study | |
| pCS2 + -Flag-RING1B | This study | |
| pCS2 + -Flag-PCGF1 | This study | |
| pCS2 + -Flag-PCGF2 | This study | |
| pCS2 + -Flag-PCGF3 | This study | |
| pCS2 + -Flag-PCGF4 | This study | |
| pCS2 + -Flag-PCGF5 | This study | |
| pCS2 + -Flag-PCGF6 | This study | |
| pCS2 + -Flag-huRYBP deletions and mutations | This study | |
| pCS2 + -Myc-CAPN3 $^{C129S}$ | Zhao et al, 2019 | |
| **Antibodies** | | |
| anti-H2AK119ub | Cell Signaling Technology | Cat# 8240 |
| anti-H3K27me3 | Cell Signaling Technology | Cat# 9733 |
| anti-H3K27me3 | Abcam | Cat# ab6002 |
| anti-H3K27ac | Abcam | Cat# ab4729 |
| anti-H3 | Beyotime | Cat# AF0009 |
| anti-Calpain 3 | Homemade | N/A |
| anti-RING1A | Abcam | Cat# ab180170 |
| anti-RYBP | Abcam | Cat# ab185971 |
| anti-RYBP | Santa Cruz Biotechnology | Cat# sc-374235 |
| anti-Tubulin | ABclonal | Cat# AC021 |
| anti-Fibrillarin | Abcam | Cat# ab4566 |
| anti-Lamin B1 | Abcam | Cat# ab133741 |
| anti-IgG | Abcam | Cat# ab171870 |
| anti-Flag | HuaBio | Cat# M1403-2 |
| anti-Myc | ABclonal | Cat# A1309 |
| anti-β-Actin | ABclonal | Cat# AC026 |
| **Oligonucleotides and other sequence-based reagents** | | |
| GGGTTTACAG TCTGCTTCCGC | This study | CAPN3 ID-KO-tF1 |
| GGAGACTGCAA GCCTGAGAAC | This study | CAPN3 ID-WT-tF1 |
| CCAGTCCTGA GAGTGAAGCG | This study | CAPN 3 ID-tR1 |
| AACGAGATTAG CCCTCTGCAGC | This study | Y70A ID-Fw |
| TCTAAGGGACAAA GCAAAGACCCAG | This study | Y70A ID-Rv |
| TCTAAGGGACAAAG CAAAGACCCAG | This study | Y70A ID-seq |
| **Chemicals, enzymes and other reagents** | | |
| DMEM High Glucose | Biological Industries | Cat# 06-1055-57-1ACS |
| Phosphate Buffered Saline | Biological Industries | Cat# 02-024-1ACS |

| Reagent/resource | Reference or source | Identifier or catalog number |
|---|---|---|
| Fetal Bovine Serum | Gibco | Cat# 10270 |
| Trypsin-EDTA | Bio-light Biotech | Cat# BL-025TE-00 |
| Williams' E medium | Gibco | Cat# 12551-032 |
| Dexamethasone | Beyotime | Cat# ST1258 |
| insulin | Beyotime | Cat# P3376 |
| Albumin Bovine Fraction V | HXbio | Cat# D332 |
| PolyJet | SignaGen | Cat# 100688 |
| cOmplete, EDTA-free | Roche | Cat# 16829800 |
| PMSF | Beyotime Biotechnology | Cat# 329-98-6 |
| DTT | MeilunBio | Cat# MB3047 |
| EcoRV restriction enzyme | New England Biolabs | Cat# R0195V |
| BamHI-HF® restriction enzyme | New England Biolabs | New England Biolabs |
| NEBuffer™ r3.1 | New England Biolabs | Cat# B6003S |
| rCutSmart™ Buffer | New England Biolabs | Cat# B6004S |
| RIPA | Beyotime Biotechnology | Cat# P0013B |
| Cell lysis buffer for Western and IP | Beyotime Biotechnology | Cat# P0013 |
| KOD OneTM PCR Master Mix | TOYOBO | Cat# KMM-101 |
| ClonExpress II One Step Cloning Kit | Vazyme | Cat# C112-01 |
| NovoNGS® CUT&Tag® 4.0 High-Sensitivity Kit for Illumina | Novoprotein | Cat# N259-YH01 |
| TRIpure Reagent | Aidlab | Cat# RN0102 |
| M-MLV Reverse Transcriptase | Invitrogen | Cat# 28025-021 |
| AceQq SYBR Green Master Mix | Vazyme | Cat# Q111-02 |
| SanPrep Column PCR Product Purification Kit | Sangon Biotech | Cat# B518141 |
| SanPrep Column Plasmid Mini-Preps Kit | Sangon Biotech | Cat# B518191 |
| EDTA, Edetic acid | Sangon Biotech | Cat# 200-449-4 |
| Calcium chloride, anhydrous | Sangon Biotech | Cat# 233-140-8 |
| Glycerol | Sangon Biotech | Cat# 231-791-2 |
| DAPI | Abcam | Cat# ab285390 |
| Alexa Fluor-488 | Abcam | Cat# ab150077 |
| Alexa Fluor-647 | Abcam | Cat# ab150115 |
| β-Mercaptoethanol | Febio science | Cat# FD2120 |
| collagenase IV | Gibco | Cat# 9001-12-1 |
| Acrylamide/Bis Soluton | Fdbio Science | Cat# FD2060 |
| ECL Buffer | Vazyme | Cat# E411-04 |
| Dynabeads™ Protein A | Invitrogen | Cat# 1001D |
| Dynabeads™ Protein G | Invitrogen | Cat# 1003D |
| anti-Flag Nanobody Magarose Beads | AlpaLifeBio | Cat# KTSM1338 |
| **Software** | | |
| Hisat2 v2.0.5 | https://www.nature.com/articles/s41587-019-0201-4 | https://github.com/DaehwanKimLab/hisat2 |

| Reagent/resource | Reference or source | Identifier or catalog number |
|---|---|---|
| DESeq2 v1.28.1 | https://genomebiology.biomedcentral.com/articles/10.1186/s13059-014-0550-8 | https://github.com/thelovelab/DESeq2 |
| Bowtie2 v2.3.5.1 | https://www.nature.com/articles/nmeth.1923 | https://github.com/BenLangmead/bowtie2 |
| samtools v1.10.2 | https://academic.oup.com/gigascience/article/10/2/giab008/6137722 | https://github.com/samtools/samtools |
| deepTools v3.5.3 | https://academic.oup.com/nar/article/44/W1/W160/2499308 | https://github.com/deeptools/deepTools |
| MACS2 v2.2.7.1 | https://genomebiology.biomedcentral.com/articles/10.1186/gb-2008-9-9-r137 | https://pypi.org/project/MACS2/2.2.7.1/ |
| Inntegrative Genomics Viewer v2.15.4 | https://www.nature.com/articles/nbt.1754 | https://www.igv.org |
| ChIPseeker v1.34.1 | https://academic.oup.com/bioinformatics/article/31/14/2382/255379 | https://github.com/YuLab-SMU/ChIPseeker |
| ChromHMM v1.24 | https://www.nature.com/articles/nmeth.1906 | https://github.com/jernst98/ChromHMM |
| BEDtools v2.28.0 | https://academic.oup.com/bioinformatics/article/26/6/841/244688 | https://github.com/arq5x/bedtools |
| ggalluvial v0.12.5 | N/A | https://CRAN.R-project.org/package=ggalluvial |
| SEACR v1.3 | https://epigeneticsandchromatin.biomedcentral.com/articles/10.1186/s13072-019-0287-4 | https://github.com/FredHutch/SEACR |
| MEGA X v10.1 | https://academic.oup.com/mbe/article/35/6/1547/4990887 | https://www.megasoftware.net |
| GraphPad Prism 9 | Dotmatics | https://www.graphpad-prism.cn |
| **Other** | | |
| Illumina NextSeq 500 | Illumina | |

## Animal strains

All animal procedures were performed in full accordance with the "Guide for the Care and Use of Laboratory Animals" issued by the Animal Ethics Committee in Zhejiang University with permission (AP CODE: ZJU20220068). Animals were housed in temperature-controlled environments with a 12-hour light-dark cycle with ad libitum access to food and water. All mouse experiments were conducted using the C57BL/6JGpt strain. *Capn3* knockout mice and *RYBP*[Y70A] knock-in mice were generated by GemPharmatech Co., Ltd. Mice were genotyped using gene specific primers listed in the Reagents and Tools Table. Sprague-Dawley rats (8 weeks old) were used in this study.

## Cell culture

293T and HepG2 were from the Cell Bank of the Chinese Academy of Sciences (Shanghai, China). Cells were cultured in Dulbecco's Modified Eagle Medium (DMEM, high glucose; Biological Industries, 06-1055-57-1ACS) containing 10% fetal bovine serum

(GIBCO, 10270) and 1×Penicillin–Streptomycin Solution (Beyotime, C0222). All cells were cultured in a standard humidified incubator at 37 °C with 5% $CO_2$.

## Tet-HA-CAPN3 and Tet-HA-CAPN3^C129S overexpression cell lines

HA-Capn3 and HA-Capn3^C129S were cloned into the pRetroX-TetOne-Puro vector (Clontech, 634309), transfected into 293 T along with the packaging plasmids VsVg and pMD-MLV at a ratio of 4:1:1. Following transfection (48 h), viral supernatants were collected twice for an additional 48 h, filtered, and applied onto HepG2 cells in the presence of polybrene (8 mg/mL). Puromycin (1 μg/mL) was added to the culture medium from day 2 to day 7 to select for stably transduced cells.

## Partial hepatectomy

In all, 2/3 partial hepatectomy was performed based on the published protocol with modifications (Mitchell and Willenbring, 2008). Briefly, two-month-old male mice were anesthetized with pentobarbital sodium (50 mg/kg, i.p.). Following a midline laparotomy of 2 cm to expose the xiphoid process, the left lateral lobe was ligated near its base using 4-0 silk thread and subsequently resected. The median lobe was then ligated and resected, leaving a small portion of tissue (~3 mm from the suprahepatic vena cava) to maintain blood outflow from right and caudate lobes (Fig. EV2A). Subsequently, the peritoneum and skin were closed, animals were placed in a clean, warm recovery area until they regained consciousness. The procedure was carried out with care to minimize pain and distress. Animals were observed closely for the first 24 h post-surgery and then twice daily for the next 3 days to monitor for any signs of complications, infection, or distress.

## Liver section immunofluorescence staining (IF)

Mice were deeply anesthetized with an overdose of pentobarbital sodium (120 mg/kg, i.p.). Liver perfusion was performed using 4% PFA, ensuring thorough fixation of tissues. The procedure was carried out with care to minimize pain and distress, and animals were monitored to confirm the absence of reflexes before proceeding with perfusion. Then livers were excised and further fixed in 4% PFA for 4 h at 4 °C. Subsequently, livers were stepwise washed stepwise in PBS, PBS-15% sucrose, and PBS-30% sucrose at 4 °C, followed by embedding in Tissue OCT compound and freezing. Tissue blocks were then cut into 6 μm sections using a cryostat (Leica CM1950). Sections on slides underwent antigen retrieval with citrate buffer (pH 6) at 96 °C for 15 min and were subsequently washed in PBS-Triton X-100 for 30 min. The slides were then blocked with PBB (PBS, 0.05% Triton X-100, 0.5% BSA)-5% goat serum for 1 h. Antibodies were diluted in PBB before being added onto the slides. After incubation with the primary antibody at 4 °C overnight, slides were washed with PBS three times (10 min each) and then incubated with corresponding secondary antibodies and DAPI diluted in PBB for 2 h, followed by three washed with PBS (10 min each). Finally, slides were mounted with 50% glycerol and images were photographed with a confocal microscope (Olympus FV1000).

## Edu injection and labeling

In all, 5-ethynyl-2′-deoxyuridine (EdU; Sigma, T511285) was dissolved in saline at 10 mg/mL as the stock solution. For EdU labeling, mice received intraperitoneal injections of EdU at a dose of 50 mg/kg every 24 h post-PHx (at 0, 24, and 48 h post-PHx). Edu signals were detected using Azide 647 labeling.

## Hepatocytes harvesting

Male mice were anesthetized with pentobarbital sodium. The livers were perfused twice: first with EGTA-D-Hank's solution (to clear blood cells and immune cells) for 5–7 min and then with collagenase IV-D-Hank's solution (for in situ digestion of the liver) for 6–8 min. After liver perfusion, the liver was dissected in DMEM medium using scissors. The homogenized liver mixture was transferred and centrifuged at 50 × g at 4 °C for 5 min. After centrifugation, the bottom pellet consisted of hepatocytes.

## Primary hepatocytes culture

Isolated primary hepatocytes were resuspended and cultured in DMEM containing 10% fetal bovine serum and 1× Penicillin–Streptomycin Solution. After 4-6 h, when the majority of the cells had adhered, the medium was replaced with Williams' E medium (GIBCO,12551-032) containing 100 nM Dexamethasone (Beyotime, ST1258) and 100 nM insulin (Beyotime, P3376).

## Liver-to-body weight ratio (LBR) detection

Mice were fasted for four hours before weight measurement. The LBR was calculated as the wet liver weight divided by the total body weight of the mouse.

## RNA extraction

Total RNA was extracted from perfused hepatocytes using TRIpureReagent (Aidlab, RN0102) according to the manufacturer's instructions. In brief, 1 mL of Trizol was used to suspend hepatocytes, and 200 μL of chloroform was added. The mixture was shaken violently for 15 s and then left at room temperature for 5 min before being centrifugation at 4 °C at 10,000 × g for 15 min. The supernatant was collected, and 500 μL isopropanol was added and gently mixed. After incubating at room temperature for 10 min, the mixture was centrifuged at 4 °C at 12,000 × g for 10 min. The supernatant was discarded, and the RNA pellet was washed twice with precooled 75% ethanol. After drying ethanol at 42 °C, DEPC water was added to dissolve the RNA.

## Bulk RNA-seq

For bulk RNA-seq, total RNA was extracted from dissected WT and *Capn3*^−/− mice hepatocytes, respectively, and the quality was assessed using the Bioanalyzer 2100 system (Agilent Technologies, CA, USA). The procedures for library construction included mRNA purification by poly-T oligo-attached magnetic beads, RNA fragmentation, reverse transcription using random hexamer primers and M-MuLV reverse transcriptase, followed by RNA degradation with RNase H treatment, second-strand cDNA

synthesis with DNA Polymerase I, blunt ends repair, 3' ends adenylation, adaptor ligation, selections of 370 ~ 420 bp cDNA fragments using the AMPure XP system (Beckman Coulter, Beverly, USA). The library was amplified with Phusion High-Fidelity DNA polymerase and quality assessment. Samples were sequenced on Illumina Novaseq platform to produce 150 bp paired-end reads. The sequencing data were deposited in the NCBI Sequence Read Archive (submission ID: SUB12531174; BioProject ID: PRJNA923397). Clean reads were mapped to the mouse genome (GRCm39) using the software Hisat2 v2.0.5 with default parameters.

## ChIP-seq

ChIP-seq were performed as described previously (Durbin et al, 2018). Hepatocytes were cross-linked in 1% formaldehyde at room temperature for 10 min, followed by 0.125 M glycine quenching for 5 min. Nucleus were extracted before sonication. Anti-H2AK119ub rabbit monoclonal (D27C4) antibody (Cell Signaling Technology, 8240), and anti-H3K27me3 rabbit monoclonal (C36B11) antibody (Cell Signaling Technology, 9733), and anti-H3K27ac rabbit polyclonal antibody (Abcam, ab4729) coupled with a mixture with Dynabeads$^{TM}$ protein A (Invitrogen,1001D) and G (Invitrogen, 1003D) were used for immunoprecipitation. ChIP DNA was purified and subjected to library preparation for Illumina NextSeq 500 Next-Generation Sequencing (Annoroad).

## CUT&Tag

CUT&Tag were performed according to the instruction of NovoNGS® CUT&Tag® 4.0 High-Sensitivity Kit (Novoprotein, N259-YH01) for fixed cells. Hepatocytes were cross-linked in 1% formaldehyde at room temperature for 10 min, followed by 0.125 M glycine quenching for 5 min. 200,000 hepatocytes were used per reaction. Anti-H2AK119ub rabbit monoclonal (D27C4) antibody (Cell Signaling Technology, 8240) were used for immunoprecipitation.

## Nuclear protein extraction and immunoprecipitation

Immunoprecipitation was performed as described previously (Shi et al, 2020). For the RING1A-IP in HepG2 cells, Ctrl, Tet-HA-CAPN3 and Tet-HA-CAPN3$^{C129S}$ stable HepG2 cells (~2 × 10$^7$ cells each) induced by 1.5 μg/mL doxycycline for 48 h were harvested and washed twice with PBS. The cells were first lysed in Buffer A (25 mM HEPES (pH 7.6), 5 mM MgCl2, 25 mM KCl, 0.05 mM EDTA, 10% Glycerol, 0.1% NP-40, 1 mM dithiothreitol (DTT), and 1× protease inhibitor cocktail (Roche, 04693132001)) to eliminate cytoplasmic protein. For RING1A-IP in rat liver, the tissue sample was homogenized in Buffer A (60 Hz, 60 s), followed by centrifugation at 2500 × g for 5 min and two washes with Buffer A to obtain the nuclear fraction. The nuclear pellets were subsequently lysed in IP lysis buffer (Beyotime, p0013) containing protease inhibitor cocktail by sonication. After centrifuging at 12,000 × g for 10 min, the supernatant was taken for immunoprecipitation. Anti-RING1A rabbit monoclonal (EPR13047) antibody (Abcam, ab180170) coupled with a mixture with Dynabeads$^{TM}$ protein A (Invitrogen, 1001D) and G (Invitrogen, 1003D) was used for immunoprecipitation. Anti-RING1A rabbit monoclonal

(EPR13047) antibody (Abcam, ab180170), anti-RYBP mouse monoclonal (A-1) antibody (Santa Cruz Biotechnology, sc-374235) and anti-LaminB1 rabbit monoclonal (EPR8985(B)) antibody (Abcam, ab133741) were used as primary antibodies for western blot analysis.

## Co-immunoprecipitations (Co-IP)

Myc-tagged CAPN3$^{C129S}$ and each FLAG-tagged PRC1 component were co-transfected into cultured human 293 T cells. After 48 h post-transfection, the cells were lysed using IP lysis buffer (Beyotime, p0013) supplemented with 1× protease inhibitor cocktail (Roche, 04693132001) and 2 mM EDTA by sonication. After centrifuging at 12,000 × g, 4 °C for 5 min, the supernatant was utilized for subsequent co-immunoprecipitations following the protocol outlined for the anti-Flag Nanobody Magarose Beads (AlpaLifeBio, KTSM1338). Western blot analysis was performed using anti-c Myc rabbit polyclonal antibody (HuaBio, 0912-2) and anti-Flag rabbit polyclonal antibodies (HuaBio, M1403-2). Other Co-IP experiments were performed using a similar protocol, except that anti-Flag Agarose-bead (Bimake, B23101) were used.

## Mass spectrometry analysis

Proteins (~30 μg) were reduced by 10 mM dithiothreitol for 30 min and alkylated by 15 mM iodoacetamide in dark for 45 min. Protein was precipitated by cold acetone centrifugation at 12,000 × g for 10 min, and then were resuspended in 50 mM ammonium bicarbonate. Trypsin (V5111, Promega) at enzyme to protein ratio of 1:50 was added for 15 h digestion at 37 °C. Peptides were then desalted by C18 spin column (Thermofisher) before LC-MSMS analysis.

Peptides were separated by a 60-min non-linear ACN gradient on a reversed-phase analytical column (Acclaim PepMap RSLC, Thermo Scientific) at a constant flow rate of 300 nL/min on an U3000 nanoUPLC system. Eluted peptides were analyzed by an Exploris 480 Orbitrap mass spectrometer (ThermoFisher Scientific). Peptides were detected in the Orbitrap at a resolution of 120,000. Peptides were selected for MS/MS using NCE setting as 30%; ion fragments were detected in the Orbitrap at a resolution of 15,000.

The acquired RAW files were loaded into MaxQuant and searched against the UniProtKB human database using the standard protocol. Trypsin with up to two missed cleavages was set. Mass tolerance of 15 ppm and 7 ppm were set for first and main search, respectively. Variable modifications were set to: oxidation(M) and acetylation (protein N-term). Carbamidomethyl(C) was set as fixed modification. Protein level 1% FDR was set to filter the result.

## In vitro CAPN3 proteolysis assay

The in vitro CAPN3 proteolysis assay was performed as previously described (Zhao et al, 2019). Cultured 293T cells expressing PRC1 components or RYBP mutations (48 h post-transfection) or CAPN3 or CAPN3$^{C129S}$ (20 h post-transfection) were rinsed twice with precooled Phosphate-Buffered Saline (PBS), followed by centrifugation at 500 × g, 4 °C for 5 min. The cells were then resuspended in the cell lysis buffer (40 mM Tris HCl-KOH, 1 mM

Dithiothreitol (DTT), 1× protease inhibitor cocktail (Roche, 04693132001), 2 mM EDTA, pH 7.4) and lysed by subjecting to liquid three times. After centrifuging at $12,000 \times g$, 4 °C for 15 min, the supernatants containing overexpressed proteins were used for further incubation.

For CAPN3 proteolysis assay, reaction mixtures were prepared by combining 50 µL of the supernatant containing candidate substrates with 50 µL of the supernatant containing CAPN3 or CAPN3$^{C129S}$, or by combining the reactions listed in the table of Fig. 3A. To each reaction, 1 µL of 500 mM CaCl$_2$ (final concentration 5 mM) was added. The reaction mixtures were then incubated at 37 °C for indicated times and were stopped by adding 4 µL of 500 mM EDTA (final concentration: 10 mM). The reaction mixtures were then denaturized at 100 °C for 10 min after adding 23.6 µL 3× protein loading buffer (120 mM Tris-HCL (PH 6.8), 30% glycerol, 12.5% β-mercaptoethanol, 3.72% SDS, 0.108 mg/mL bromophenol blue, 2 mM EDTA, 1× protease inhibitor cocktail (Roche, 04693132001)) and subjected to SDS-PAGE for western blot analysis.

## Cell immunofluorescence staining

Immunofluorescence staining of the cultured cells was as previously described (Zhao et al, 2019). Cells were plated on glass slides and subsequently fixed using 4% paraformaldehyde for 10 min. Subsequently, cells were permeabilized using PBS-Triton (0.2% Triton X-100 in PBS). Samples were washed in PBS once and blocked with blocking solution (90% PBS, 5% goat serum, 5% fetal bovine serum, 2% newborn calf serum) for 30 min at room temperature. Primary antibodies incubation with anti-RYBP (dilution 1:100), anti-Fibrillarin (dilution 1:100), anti-H2AK119ub (dilution 1:800), anti-H3K27me3 (dilution 1:200) was performed in blocking solution overnight at 4 °C, followed by three 5-min washes with PBS-Triton. Secondary antibodies (dilution 1:300) and DAPI (dilution 1:300) were added, and samples were incubated for a further 1 h at room temperature. After three quick washes with PBS-Triton, the glass slides were finally mounted in 80% glycerol and images were acquired using a laser confocal microscope (Olympus FV1000).

## Chromatin fraction isolation

Chromatin fractions were isolated as described previously (Xu et al, 2018). Cells were harvested and washed twice with PBS. The cells were firstly lysed in CSK buffer (10 mM PIPES, pH 6.8, 300 mM sucrose, 100 mM NaCl, 1 mM MgCl$_2$, 1 mM EDTA, 1 mM EGTA, 0.5% Triton X-100) supplemented with 1 mM dithiothreitol (DTT), 1 mM phenylmethylsulfonyl fluoride (PMSF), 1× protease inhibitor cocktail. After centrifugation, chromatin pellets were washed twice with CSK buffer, resuspended in lysis buffer (50 mM Tris pH 8.0, 420 mM NaCl, 5% glycerol, 0.1% NP-40, 0.1 mM EDTA) with freshly added 1 mM DTT, 1 mM PMSF, 1× protease inhibitor cocktail (Roche), 1 mM MgCl$_2$ and Benzonase nuclease (1:500, Novagen), vortexed, and digested with Benzonase nuclease for 30 min on ice. Clear supernatant after centrifugation (13,000 rpm, 10 min) represented the chromatin fraction for subsequent analysis. Bradford assay was used for protein quantification.

## Differential gene-expression analysis of RNA sequencing data

For the bulk RNA-seq, after mapping the clean reads to the mouse genome (GRCm39) using the software Hisat2 (v2.0.5) (Kim et al, 2019) with default parameters, transcript levels were obtained for each gene and the differentially expressed genes (DEGs) was determined with DESeq2 (v1.28.1) (Love et al, 2014).

## ChIP-seq data analysis

Adaptors and low-quality reads were removed through Trim Galore (v0.6.4) using the parameter '--paired'. Trimmed reads were then aligned to the mouse reference genome GRCm39/mm39 using Bowtie2 (v2.3.5.1) (Langmead and Salzberg, 2012). with the parameter '--very-sensitive-local --no-unal --no-mixed --no-discordant -I 10 -X 700'. Bam files were generated, sorted, and indexed using samtools (v1.10.2) (Danecek et al, 2021) with default parameters. Picard (v3.1.0) was used to mark the PCR duplicates with the parameter '-REMOVE_DUPLICATES false'. Following this, 'bamCoverage' command from deepTools (v3.5.3) (Ramírez et al, 2016) was used to convert bam files to bigwig files with a bin size of 10. Peaks were called by MACS2 (v2.2.7.1) with $P$ value of 10e-6.

To visualize ChIP-seq signals, functions computeMatrix, plotHeatmap and plotProfile from deepTools was used to draw heatmaps and profile diagrams. Normalized signals were visualized in Inntegrative Genomics Viewer (IGV) (v2.15.4). Peak distribution and annotation were generated by R packages ChIPseeker (v1.34.1) (Yu et al, 2015).

The ChromHMM (v1.24) (Ernst and Kellis, 2012) was used to define chromatin states based on H2AK119ub H3K27me3 and H3K27ac signal distributions throughout the genome. To be specific, bam files were first binarized using 'BinarizeBam' command with a bin size of 5 kb and a signal-to-background fold enrichment greater than two. The segmentation model was then trained using the 'LearnModel' command with a bin size of 5 kb and a maximum of 300 iterations. Finally, genomic regions were classified into four chromatin states using command of 'CompareModel'. The median TPM of transcriptional levels correspond to each chromatin states were calculated and plotted for each sample. To make the alluvial diagram, BEDtools (v2.28.0) was used to cut the genome with a window size of 5 kb, and a R package ggalluvial (v0.12.5) was applied to plot the diagram.

## Model-based clustering

Covariance clustering was performed using the R package Mclust (Scrucca et al, 2016). Mean TPM values across three biological replicates per conditions of the RNA-seq data were converted to z-scores. Genes with mean TPM > 0.1 in ≥1 condition were retained. Bayesian information content (BIC) was computed with mclustBIC (modelNames = "VVI", G = seq (2,30,2)) to assess 2–30 component models, and the first stepwise increase reducing BIC determined the optimal cluster number (16). Clustering was performed with Mclust (data = 16, modelNames = "VVI", prior = priorControl()).

## Phylogenetic tree construction

The amino acids sequences of RYBP or its homologous proteins harbored in a series of species were collected from National Center for Biotechnology Information (NCBI). Accession numbers of these sequences were NP_036366.3[Homo sapiens], XP_030865299.1[Gorilla], XP_015202772.1[Lepisosteus oculatus], XP_051707435.1[Oryctolagus cuniculus], NP_001386529.1[Rattus norvegicus], NP_062717.2[Mus musculus], NP_001016846.1[Xenopus tropicalis], XP_039582624.1[Passer montanus], XP_039189423.1[Crotalus tigris], NP_958474.2[Danio rerio], XP_042195565.1[Callorhinchus milii], XP_032819297.1[Petromyzon marinus], XP_026690924.1[Ciona intestinalis], XP_038049344.1[Patiria miniata], XP_780545.2[Strongylocentrotus purpuratus], XP_033096921.1[Anneissia japonica], KAI8481789.1[Branchiostoma belcheri], NP_611705.1[Drosophila melanogaster]Global alignment of amino acids sequences was conducted using a web tool 'Clustal W' with default parameters. Maximum likelihood (ML) phylogenetic tree was constructed using MEGA X (v10.1).

## Statistical analysis

Unless stated otherwise, all parameters were tested using unpaired two-tailed Student's *t* test. Significant *P* value in all statistical analyses was obtained using GraphPad Prism 9 (GraphPad Software). A *P* value below 0.05 was considered to be statistically significant.

# Data availability

All RNA-seq and ChIP-seq data has been deposited in the GEO database (SRA: PRJNA1040491).

The source data of this paper are collected in the following database record: biostudies:S-SCDT-10_1038-S44318-026-00729-9.

# Peer review information

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

## Acknowledgements

We would like to thank Dr. Ye Chen, Dr. Yun Bai, Dr. Yong Wang and all the members of the Hui Shi, Jinrong Peng, Jun Chen, Li Jan Lo and Zhipeng Ma labs for their generous assistance and valuable suggestions. This work is financially supported by the National Key R&D Program of China (2023YFA1800602 and 2018YFA0800502), National Natural Science Foundation of China (32100658 and U21A20198). HS is also supported by Hangzhou Chengxi Sci-tech innovation Corridor Management Committee. SZY is supported by Shandong Provincial Laboratory Project (SYS202202) and Research Project of Jinan Microecological Biomedicine Shandong Laboratory (JNL-2025002B).

## Author contributions

**Wei Cui**: Conceptualization; Data curation; Formal analysis; Validation; Investigation; Visualization; Methodology; Writing—original draft; Writing—review and editing. **Qingyang Li**: Conceptualization; Data curation; Formal analysis; Validation; Investigation; Visualization; Methodology; Writing—original draft; Writing—review and editing. **Jinsong Wei**: Conceptualization; Data curation; Formal analysis; Validation; Investigation; Visualization; Methodology; Writing—original draft; Writing—review and editing. **Ting Zhou**: Conceptualization; Data curation; Formal analysis; Validation; Investigation; Visualization; Methodology; Writing—review and editing. **Haozhe Zhu**: Data curation; Formal analysis; Validation; Investigation; Visualization; Writing—original draft; Writing—review and editing. **Delai Huang**: Investigation. **Shuai Wang**: Investigation. **Jianan Gao**: Investigation. **Ru Zhou**: Investigation. **Zeyu Sun**: Investigation. **Hua Ruan**: Investigation. **Li Jan Lo**: Supervision; Writing—review and editing. **Ting Tao**: Formal analysis; Writing—review and editing. **Jun Chen**: Supervision; Writing—review and editing. **Jinrong Peng**: Supervision; Funding acquisition; Project administration; Writing—review and editing. **Hui Shi**: Conceptualization; Resources; Data curation; Formal analysis; Supervision; Funding acquisition; Validation; Investigation; Visualization; Methodology; Writing—original draft; Project administration; Writing—review and editing.

Source data underlying figure panels in this paper may have individual authorship assigned. Where available, figure panel/source data authorship is listed in the following database record: biostudies:S-SCDT-10_1038-S44318-026-00729-9.

## Disclosure and competing interests statement

The authors declare no competing interests.

# Expanded View Figures

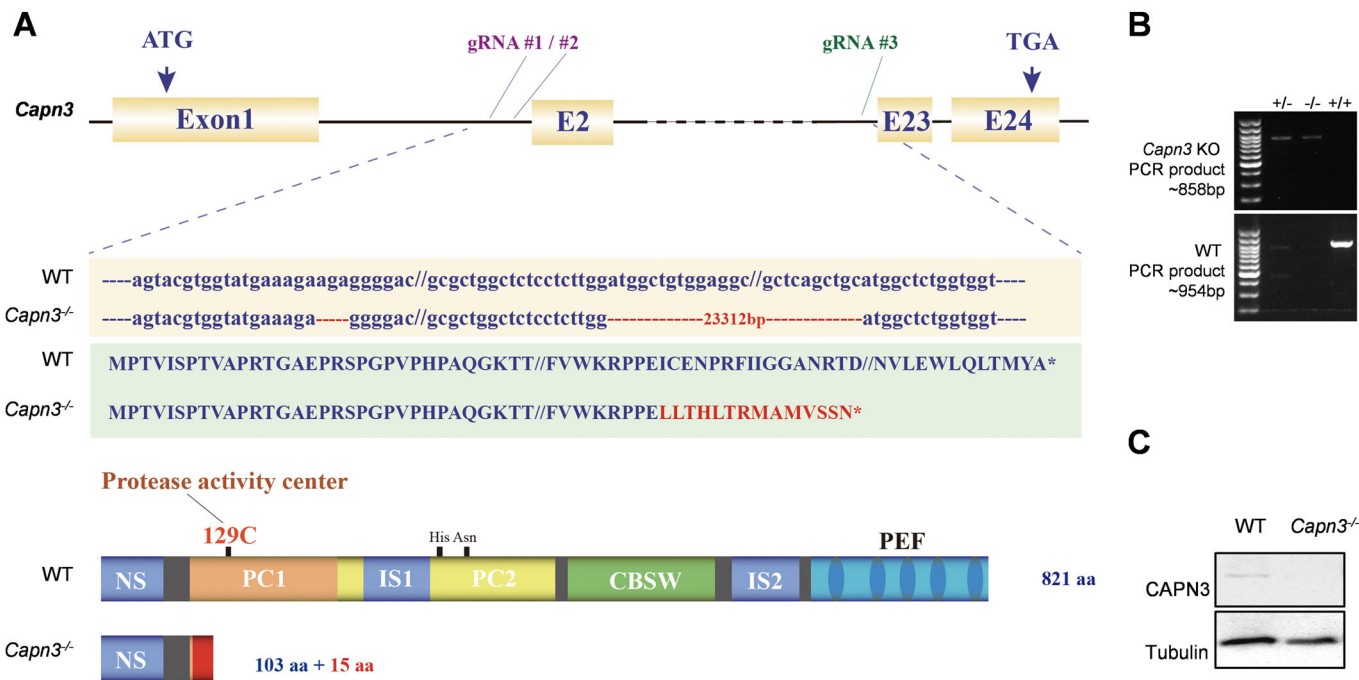

**Figure EV1. Generation of *Capn3* KO mice by CRISPR/Cas9 genome editing system.**

(A) Scheme of gRNA target sites for generating *Capn3* KO mice and the alignments of nucleotide sequences (top), protein sequence (middle), and domain (bottom) from wild-type (WT) and *Capn3*⁻/⁻ mice. (B) *Capn3*⁻/⁻ confirmed by PCR genotyping. The first pair of primers can only detect the KO product about 858 bp in *Capn3*⁺/⁻ and *Capn3*⁻/⁻ (top), the second pair of primers can only detect the WT product about 954 bp in *Capn3*⁺/⁻ and WT (bottom). (C) Western blot analysis of CAPN3 protein in WT and *Capn3*⁻/⁻ hepatocytes, which were isolated by perfusion from the WT and *Capn3*⁻/⁻ mice livers.

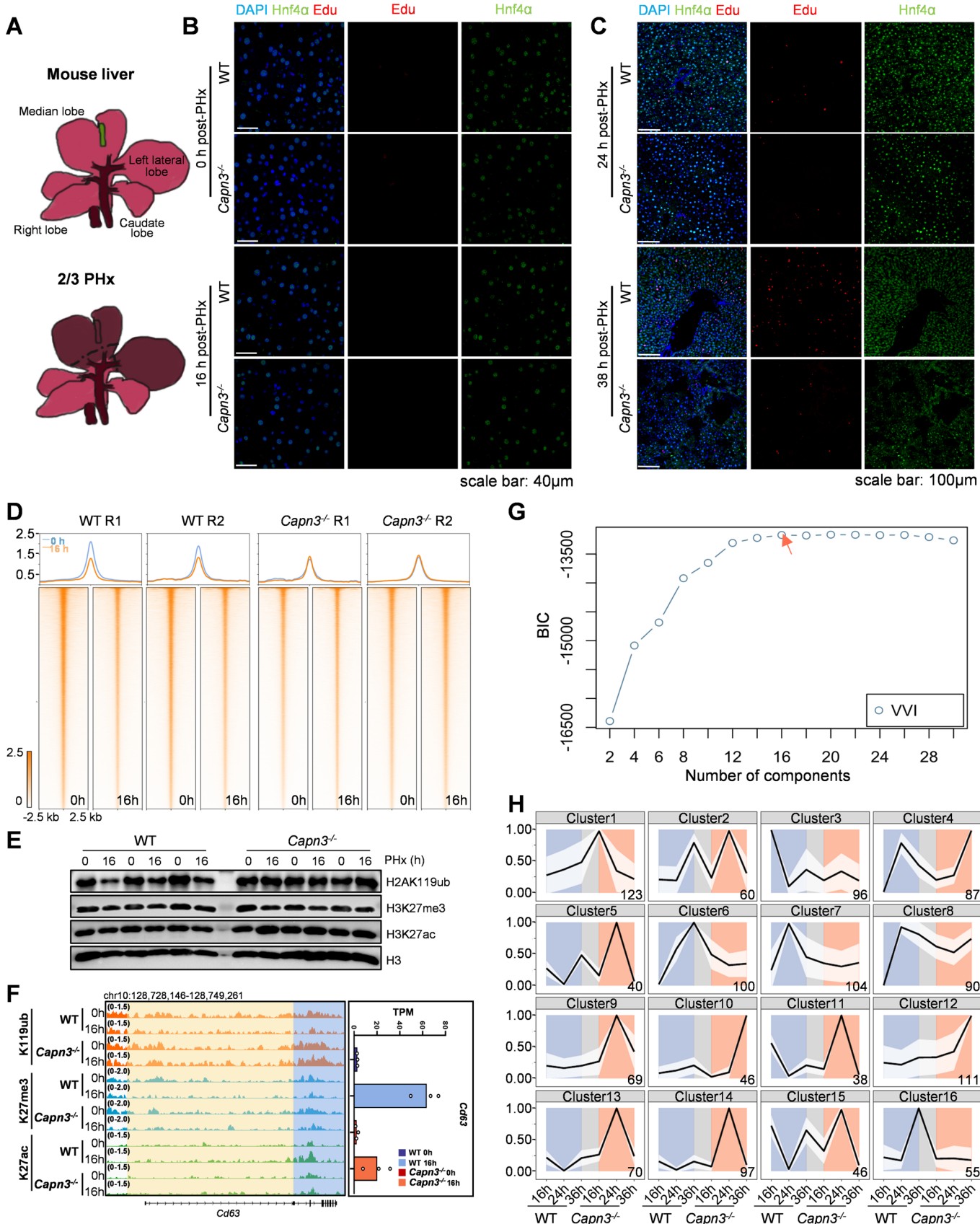

◄ **Figure EV2.   Depletion of CAPN3 delays hepatocyte proliferation and inhibits H2AK119ub remodeling after PHx.**

(A) Scheme of mouse liver anatomy and positioning of silk threads for knots (dash lines). (B) Immunostaining of EdU and Hnf4a (a hepatocyte marker) at 0 h and 16 h post-PHx. Scale bars, 40 μm. (C) Immunostaining of EdU and Hnf4a at 24 h and 38 h post-PHx. Scale bars, 100 μm. (D) Two independent biological replicates of CUT&Tag profiles and heatmaps of H2AK119ub in WT and *Capn3*$^{-/-}$ hepatocytes at 0 h and 16 h post-PHx at called peaks. Color intensity for each strand represents counts per million (CPM). (E) Chromatin fractions were isolated from the livers of three independent WT and *Capn3*$^{-/-}$ mouse. Western blot analysis was performed with the indicated antibodies. (F) Genome browser snapshots (left) of H2AK119ub, H3K27me3, and H3K27ac tracks at the *Cd63* loci in WT and *Capn3*$^{-/-}$ hepatocytes at 0 h or 16 h post-PHx. The expression levels (right) of *Cd63* in TPM at 0 h or 16 h post-PHx are summarized in bar charts. The shaded areas highlight the #41_44 (blue) and #42_44 (yellow) regions. (G) Bayesian information content (BIC) was computed to assess 2–30 component models to get the optimal cluster number (indicated by orange arrow). (H) Covariance clustering of genes near #41_44, #42_44 and #43_44 loci that display a delayed reduction in H2AK119ub (TPM > 0.1 in ≥1 condition, $n = 1232$) across 16, 24, and 36 h PHx in WT and *Capn3*$^{-/-}$ hepatocytes into 16 clusters. Gene counts for each cluster are shown at the bottom right of each corresponding panel. Source data are available online for this figure.

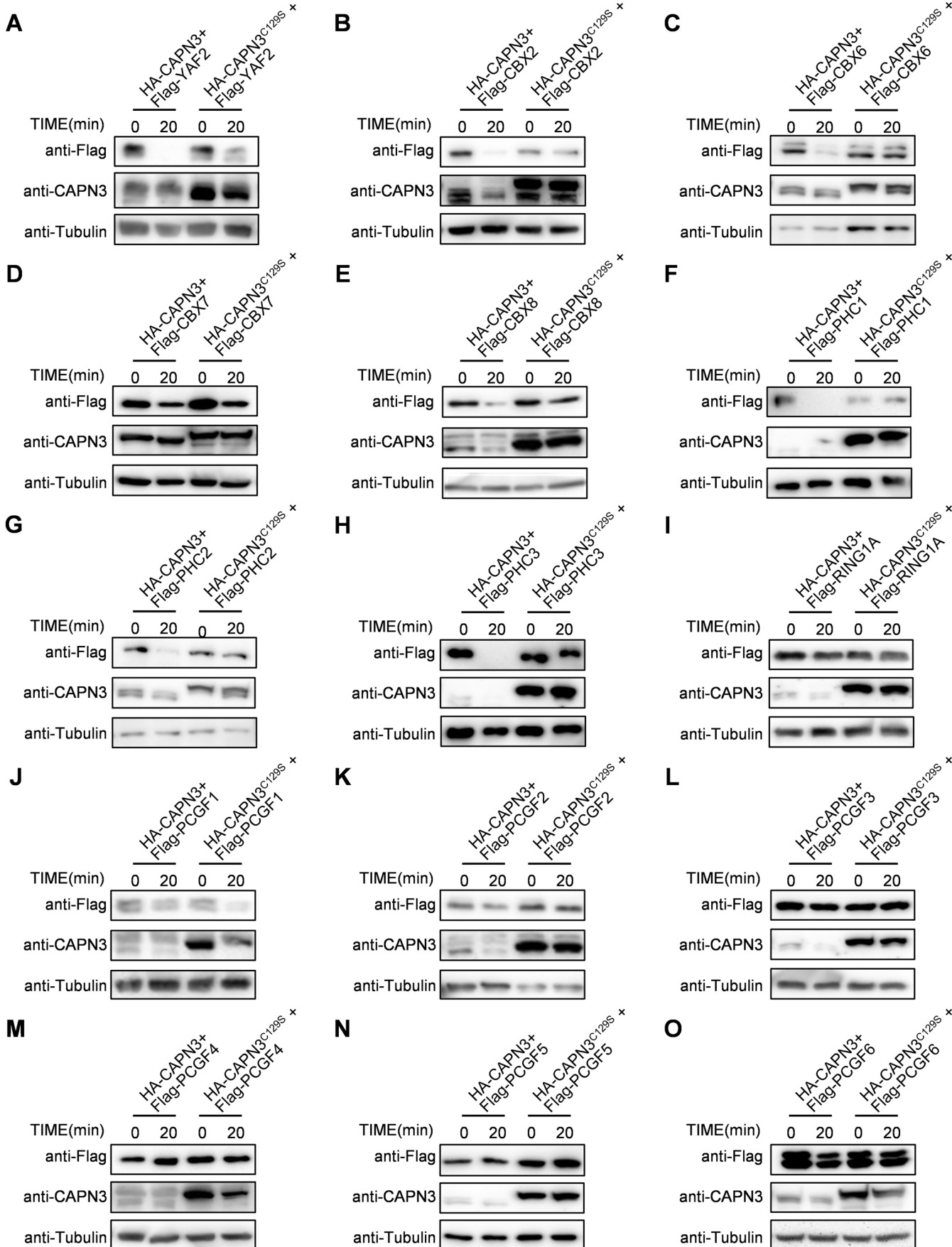

**Figure EV3. CAPN3-mediated proteolysis of PRC1 components.**

(A–O) Western blot analysis of in vitro CAPN3 proteolysis assays of PRC1 components (A) YAF2, (B) CBX2, (C) CBX6, (D) CBX7, (E) CBX8, (F) PHC1, (G) PHC2, (H) PHC3, (I) RING1A, (J) PCGF1, (K) PCGF2, (L) PCGF3, (M) PCGF4, (N) PCGF5, (O) PCGF6. The incubation times and antibodies used are indicated. These experiments were replicated at least twice.

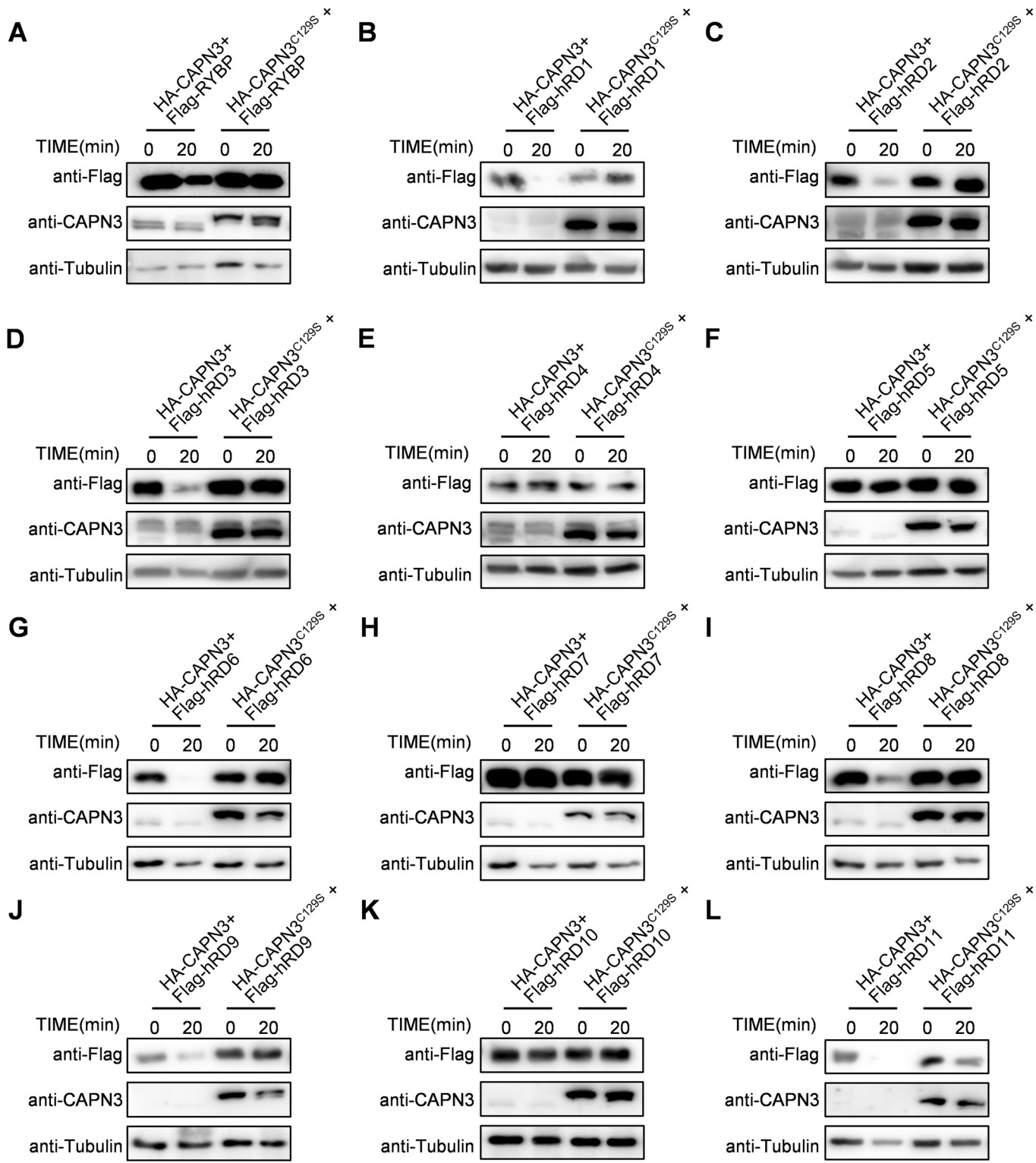

**Figure EV4. The 67th to 75th amino acids of RYBP are required for the proteolysis of RYBP by CAPN3.**

(A–L) Western blot analysis of in vitro CAPN3 proteolysis assays of different amino acids deletion mutants of human RYBP. (A) RYBP, (B) hRD1, (C) hRD2, (D) hRD3, (E) hRD4, (F) hRD5, (G) hRD6, (H) hRD7, (I) hRD8, (J) hRD9, (K) hRD10, (L) hRD11. The incubation times and antibodies used are indicated. These experiments were replicated at least twice.

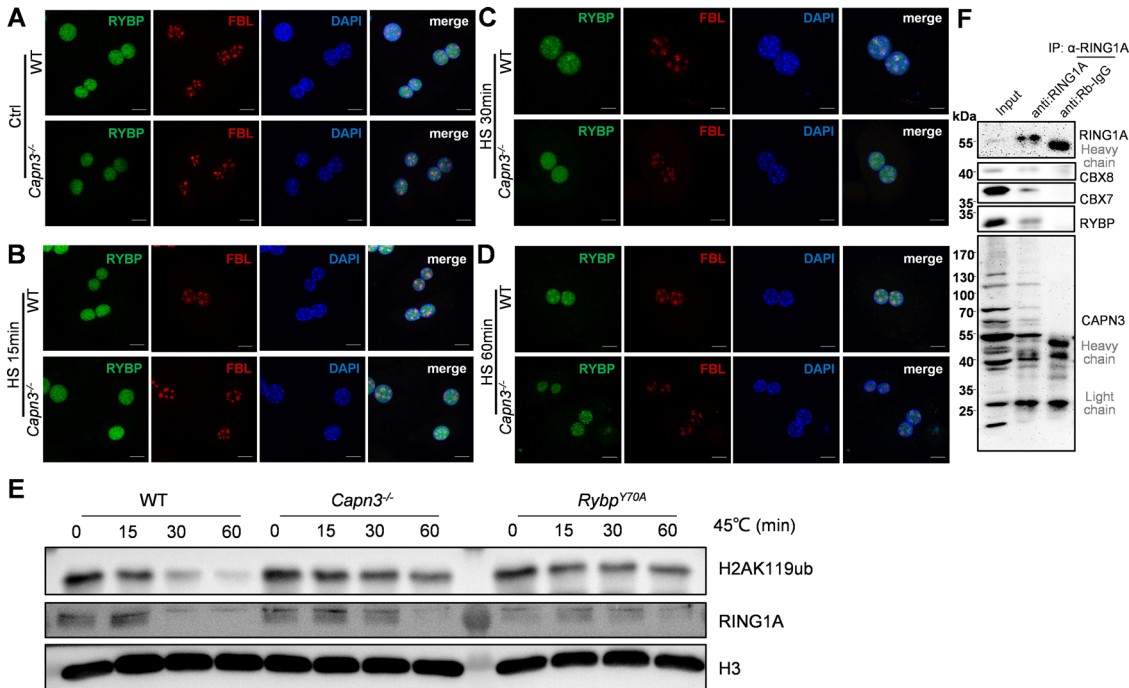

**Figure EV5.  CAPN3 facilitates nucleolus entry of RYBP and H2AK119ub elimination during stress response.**

(A–D) Immunostaining of RYBP and FBL (Fibrillarin) (a nucleolus marker) in WT and *Capn3⁻/⁻* primary hepatocyte cells after incubated at 45 °C for 0, 15, 30, or 60 min. Scale bars, 10 µm. (E) Chromatin fractions were isolated from WT, *RYBP^Y70A* homozygous and *Capn3⁻/⁻* primary hepatocytes incubated at 45 °C for 0, 15, 30, or 60 min. Western blot analysis was performed with indicated antibodies. This experiment was replicated at least twice. (F) Western blot analysis of Input and immunoprecipitation samples from rat liver nuclear protein extracts (without PHx or heat shock). 'Input' contains ~10% of the input cell lysate used for IP. Antibodies used are indicated.

