## [Peer Review File · The EMBO Journal]

Protease-mediated PRC1 dissociation promotes H2AK119ub remodeling during stress responses

Wei Cui, Qingyang Li, Jinsong Wei, Ting Zhou, Haozhe Zhu, Delai Huang, Shuai Wang, Jianan Gao, Ru Zhou, Zeyu Sun, Hua Ruan, Li Jan Lo, Ting Tao, Jun Chen, Jinrong Peng, and Hui Shi

Corresponding author(s): Hui Shi (hui_shi@zju.edu.cn)

Review Timeline:

Submission Date:	7th Jun 24
Editorial Decision:	4th Jul 24
Appeal Received:	5th Jul 24
Editorial Decision:	27th Sep 24
Appeal Received:	14th Jan 25
Editorial Decision:	21st Mar 25
Revision Received:	19th Sep 25
Editorial Decision:	25th Nov 25
Revision Received:	16th Dec 25
Accepted:	20th Jan 26

Editor: Cornelius Schneider

Transaction Report:

Dear Dr. Shi,

Thank you for submitting your manuscript "Protease-mediated PRC1 dissociation promotes H2AK119ub remodeling during stress response" to The EMBO Journal. I have now carefully read your study and discussed the manuscript with other members of the editorial team. I have also consulted with an advisor expert in this field of research. After all these considerations I regret to inform you that we have decided not to pursue the publication at The EMBO Journal.

We appreciate that your extensive and detailed manuscript describes a new mechanism to regulate PRC1-mediate repression upon stress. The manuscript proposes that de-repression of a subset of genes is mediate by recruitment of the protease CAPN3 to core PRC1 subunits and the subsequent degradation of peripheral subunits which should then lead to the dissociation of PRC1 and subsequent de-ubiquitination of H2AK119ub. However, we do not think that there is sufficiently strong evidence both for the recruitment of CAPN3 to core PRC1 subunits and there is also no information how this is regulated. In addition, we are also concerned that there is a strong variation in the in vitro cleavage assays (self-cleavage of CAPN3 in figures 2/EV2).

Our external advisor shares these concerns. As a side note this advisor also remarks that there is a lack of detail in the materials and methods section regarding precautions for animal welfare during surgery/recovery or intracardial perfusion with PFA. Taken together we therefore do not think that the advance provided here is sufficient for publication at the EMBO Journal.

Thank you for giving us the opportunity to consider this manuscript. I am very sorry for the very unusual delay in reading your manuscript which was caused by exceptional circumstances beyond my control. I hope that this will not preclude you from submitting your research to our journal in the future.

Yours sincerely,

Cornelius

Cornelius Schneider, PhD
Editor
The EMBO Journal
c.schneider@embojournal.org

** As a service to authors, EMBO Press provides authors with the possibility to transfer a manuscript that one journal cannot offer to publish to another EMBO publication or the open access journal Life Science Alliance launched in partnership between EMBO Press, Rockefeller University Press and Cold Spring Harbor Laboratory Press. The full manuscript and if applicable, reviewers' reports, are automatically sent to the receiving journal to allow for fast handling and a prompt decision on your manuscript. For more details of this service, and to transfer your manuscript please click on Link Not Available. **

Dear Dr. Schneider,

We sincerely appreciate the time and effort you and your colleagues have invested in evaluating our manuscript "Protease-mediated PRC1 dissociation promotes H2AK119ub remodeling during stress response". We fully understand your concerns due to our unclear statement. We have put more data here and clarified each of your concerns. A point-by-point response is provided below:

Q1: *"The manuscript proposes that de-repression of a subset of genes is mediate by recruitment of the protease CAPN3 to core PRC1 subunits and the subsequent degradation of peripheral subunits which should then lead to the dissociation of PRC1 and subsequent de-ubiquitination of H2AK119ub. However, we do not think that there is sufficiently strong evidence both for the recruitment of CAPN3 to core PRC1 subunits and there is also no information how this is regulated."*

A1: We appreciate your concern regarding the data we presented in the manuscript. Firstly, we apologize for any lack of clarity in our original statement. The recruitment of CAPN3 to core PRC1 subunits is due to their high affinity. The activity of CAPN3 is regulated by stress signals, most likely through the modulation of Ca²⁺ levels as CAPN3 acts as a calcium-activated protease.

We have monitored the intracellular Ca²⁺ levels of HepG2 cells following incubation at 45 °C using Fluo-4 AM (Beyotime, S1060). The results indicated that Ca²⁺ concentration, which can activate CAPN3 activity, can be elevated by heat shock. These new data are shown below:

Additionally, we realized the need for more data on the endogenous interaction of RING1A and CAPN3. Unfortunately, the antibodies of RING1A for immunoprecipitation do not work

well for mice. Therefore, we employed rat liver tissue for immunoprecipitation using an anti-RING1A antibody. We successfully detected a series endogenous CAPN3 bands pulled by endogenous RING1A compared to Rb-IgG control, indicating the endogenous interaction of these two proteins in the rat liver. The interaction of RING1A and other PRC1 components in rat liver are also verified in this experiment. These new data are shown below:

Q2: “In addition, we are also concerned that there is a strong variation in the *in vitro* cleavage assays (self-cleavage of CAPN3 in figures 2/EV2).”

A2: We apologize for the unclear statement in our manuscript. The figures in question were not collected in a single batch, which may have led to the observed variation in CAPN3 levels. Typically, we analyze 3-6 potential substrates per batch, including a negative control resistant to CAPN3 proteolysis and a positive control susceptible to it, to ensure consistency and reliability. Each potential substrate underwent testing at least twice. If you still have concerns, we are prepared to organize the results by batches for your review or analyze the substrates in batches as per your preferences. Below is an example of Western blot analysis from a single batch showing the similar CAPN3 levels:

Q3: *“Our external advisor shares these concerns. As a side note this advisor also remarks that there is a lack of detail in the materials and methods section regarding precautions for animal welfare during surgery/recovery or intracardial perfusion with PFA.”*

A3: We sincerely apologize for the oversight regarding missing the details on animal welfare during surgery/recovery or intracardial perfusion with PFA. We have revised the paragraph in the materials and methods section to include detailed precautions taken to ensure animal welfare.

The revised texts are included below:

“Partial hepatectomy

Mice were anesthetized with pentobarbital sodium (50 mg/kg, i. p.), 2/3 hepatectomy (removing the middle lobe and left lateral lobe) was performed based on the protocol as described previously (Mitchell and Willenbring, 2008). The procedure was carried out with care to minimize pain and distress, animals were placed in a clean, warm recovery area until they regained consciousness. They were monitored continuously for signs of pain, distress, and proper recovery from anesthesia. Animals were observed closely for the first 24 hours post-surgery and then twice daily for the next three days to monitor for any signs of complications, infection, or distress.”

“Liver section Immunofluorescence staining (IF)

Mice were deeply anesthetized with an overdose of pentobarbital sodium (120 mg/kg, i. p.). Intracardial perfusion was performed using 4% PFA, ensuring thorough fixation of tissues. The procedure was carried out with care to minimize pain and distress, and animals were monitored to confirm the absence of reflexes before proceeding with perfusion. Then livers were excised and further fixed in 4% PFA for 4 hours at 4°C.”

Please let me know if these additional data and clarifications address your concerns and if you would be willing to reconsider our manuscript for further steps towards publication in The EMBO Journal. Our study offers novel insights into the regulation of PRC1-mediated repression under stress conditions, which we believe is of significant interest to the readership of The EMBO Journal. If acceptable, we will integrate these data and resubmit the manuscript.

Thank you for your time and consideration. We look forward to your response.

Sincerely,

Hui

Hui Shi, Ph.D.

ZJU100 Young Professor

MOE Key Laboratory of Biosystems Homeostasis & Protection

College of Animal Sciences

Zhejiang University

866 Yuhangtang Road

Hangzhou 310058, China

Hui_shi@zju.edu.cn

Dear Dr. Shi,

Thank you for submitting your manuscript (EMBOJ-2024-118128R-Q) to The EMBO Journal. Thank you also for sharing additional data with us. We did not receive any updates for an extended period of time after our last message which you can find below. We will therefore not consider your appeal further.

With best regards,

Cornelius Schneider

Cornelius Schneider, PhD
Editor | The EMBO Journal
c.schneider@embojournal.org

** As a service to authors, EMBO Press provides authors with the possibility to transfer a manuscript that one journal cannot offer to publish to another EMBO publication or the open access journal Life Science Alliance launched in partnership between EMBO Press, Rockefeller University Press and Cold Spring Harbor Laboratory Press. The full manuscript and if applicable, reviewers' reports, are automatically sent to the receiving journal to allow for fast handling and a prompt decision on your manuscript. For more details of this service, and to transfer your manuscript please click on Link Not Available. **

Dear Hui,

Thank you very much for sharing this additional data with us. In general, I think that the additional information that you have provided is helpful and informative.

I have one additional question regarding the interaction between CAPN3 and RING1A. Is this interaction induced by stress or is CAPN3 always bound to PRC1, and the activity is controlled mostly by Ca⁺⁺ as you mention in your letter? I am still somewhat concerned that the apparent somewhat variable self-degradation by CAPN3 in the in vitro cleavage experiments makes the comparison between the different PRC1 subunits difficult and it would be therefore helpful to show at least 2 replicates of each in vitro cleavage assay for all tested proteins. Thank you also for providing additional information regarding the mouse experiments. The advisor also mentioned that "partial hepatectomies are major operations and the quality of surgery/recovery will have an impact on their data" and we would therefore ask you to provide a detailed protocol of the surgery and a supplementary figure which shows the quality control measures that you have taken in controlling the quality and reproducibility of this experiment.

In summary, I would offer to send this manuscript for peer review if you
provide evidence for the recruitment and regulation of CAPN3 as mentioned above
include the replicates for the in vitro cleavage experiments
add the requested additional detail to the materials and methods and quality control section of the manuscript.

With best regards,

Cornelius Schneider

Cornelius Schneider, PhD
Editor | The EMBO Journal
c.schneider@embojournal.org

Dear Dr. Schneider,

Happy new year! I hope this message finds you well.

I sincerely apologize for my delayed response. I was recently involved in a car accident, which caused some unexpected disruptions. I truly appreciated your understanding and patience. During this period, we have acquired additional *RYBP^{Y70A}* knock-in mice and performed more experiments. The new data we obtained has further strengthened our conclusions, and we believe it will address the concerns you raised. These updates have been incorporated into the revised manuscript. For your convenience, we first list all relevant new data below:

1. **Figure 1J**: A rearranged heatmap of RNA-Seq data showing a clearer dynamic of gene expressions.
2. **Figure 5A**: A revised schematic illustrating the strategy for constructing the *RYBP^{Y70A}* knock-in mice with additional details.
3. **Figure 5C**: An expanded LBR analysis of *RYBP^{Y70A}* knock-in mice, incorporating additional replications and time points.
4. **Figure 6D**: A new Western blot analysis of the *in vitro* cleavage assay for RYBP, both in the presence and absence of free calcium ions, showing the cleavage of RYBP by CAPN3 is Ca⁺⁺ dependent.
5. **Figure 6E**: A new Western blot analysis of the chromatin fractions isolated from wild-type and *Capn3^{-/-}* primary hepatocytes, with/without 20 μM BAPTA (a Ca⁺⁺ chelator) pretreated for 1h, followed by incubation at 45°C for 0, 15, 30 or 60 min, showing the heat shock induced CAPN3-PRC1 pathway is Ca⁺⁺ dependent.
6. **Figure EV1D**: A new schematic illustrating the mouse liver anatomy and the detailed positioning of silk threads for knots.
7. **Figure EV4E**: A new Western blot analysis showing the delayed reduction of H2AK119ub in both homozygous *RYBP^{Y70A}* knock-in mice and *Capn3^{-/-}* mice derived primary hepatocytes following heat shock treatment.
8. **Figure EV4F**: The Western blot analysis of the immunoprecipitation data, as we mentioned in our previous email, showing the interaction between CAPN3 and RING1A in normal rat liver.
9. **Repeat of Figures 2 D and E, 4 B and C, EV2 and EV4**: Repeated analyses of each *in vitro* CAPN3 proteolysis are provided below.

.....

Based on these new data together with the data presented in our previous submission, we have provided a point-by-point response to your concerns below.

Q1: *I have one additional question regarding the interaction between CAPN3 and RING1A. Is this interaction induced by stress or is CAPN3 always bound to PRC1, and the activity is controlled mostly by Ca⁺⁺ as you mention in your letter?*

A1: We apologize for the unclear statement in our earlier appeal. To clarify, the rat liver used for immunoprecipitation in our study was obtained from a rat that had not undergone partial hepatectomy or heat shock (**Fig. EV4F**). Therefore, the binding of CAPN3 to PRC1 occurred under normal conditions, indicating that the interaction is not induced by stress.

CAPN3 has been identified as a calcium-dependent cysteine protease. Here, we further validated the cleavage of RYBP by CAPN3 is also depended on the presence of Ca⁺⁺ (**Fig.6 D**). Additionally, to determine whether the facilitation of H2AK119ub elimination by CAPN3 under stress conditions is also Ca⁺⁺ dependent, we pre-treated cells with the Ca⁺⁺ chelator BAPTA and incubated them at 45 °C for 0, 15, 30 and 60 minutes. Chromatin was subsequently extracted and western blot analysis was performed. The results revealed BAPTA treatment delayed the elimination of both H2AK119ub and CBX8 in wild-type cells, but not in KO cells, during heat shock, suggesting that the facilitation of this process by CAPN3 is Ca⁺⁺ dependent (**Fig.6E**).

These new data are now shown as Fig. EV4F, Fig. 6D and E and described in the text of the paper.

Page 9, paragraph 3:

“We then sought to distinguish whether stress signals trigger the recruitment of CAPN3 to the PRC1 complex or activation of CAPN3 in the PRC1 complexes to compromise the chromatin-bound PRC1. Due to the ineffectiveness of the RING1A antibody in mice, we had to conduct an immunoprecipitation using a rat liver that was not subjected to any stress treatment, including PHx or heat shock. This assay revealed that the endogenous CAPN3 is capable of binding with PRC1 complexes prior to the imposing of stress signals (Fig. EV4F). CAPN3 is a calcium-dependent cysteine protease. We then performed the *in vitro* cleavage assay for RYBP in both the presence and absence of free calcium ions. The results indicated that RYBP was cleaved by CAPN3 only in the presence of free calcium ions, demonstrating that this reaction is calcium ion-dependent (Fig. 6D). Thus, we hypothesized that CAPN3 binds to PRC1 complexes prior to a response to stress signals and is likely activated by free calcium ions induced by cellular stresses. To test this hypothesis, we pre-treated hepatocytes with the calcium chelator BAPTA and incubated them at 45 °C for 0, 15, 30 and 60 minutes. Chromatin was extracted and western blot analysis was performed. The results indicated that BAPTA treatment delayed the elimination of both H2AK119ub and CBX8 from chromatin in wild-type hepatocytes during heat shock, but not in *Capn3*^{-/-} samples, suggesting that stress induced calcium fluctuation might play a key role in activating the CAPN3-PRC1 regulatory pathway (Fig. 6E).”

Page 11, paragraph 2 (Discussion):

“In addition, we found that chelating calcium attenuated the effect of heat shock on reducing the H2AK119ub level. Considering the fact that CAPN3 can complex with the PRC1, we propose that calcium is an activator of the CAPN3-PRC1 pathway upon stress signals such as PHx and heat shock.”

Figure EV4F and legend:

“

Figure EV4. CAPN3 facilitates nucleolus entry of RYBP and H2AK119ub elimination during stress response.

(F) Western blot analysis of Input and immunoprecipitation samples from rat liver nuclear protein extracts (without PHx or heat shock). Antibodies used are indicated.”

Figure 6 D and E and legend:

“

Figure 6. Proteolysis of PRC1 non-core components by CAPN3 disrupts the maintenance of genomic H2AK119ub by PRC1 occupation

(D) Western blot analysis of *in vitro* CAPN3 proteolysis assay of human RYBP. Reaction mixture (containing 2 mM EDTA, chelated endogenous Ca²⁺) was incubated at 37°C for 10 min with or without 2.5 mM Ca²⁺ as indicated.

(E) Chromatin fractions were isolated from WT and *Capn3*^{-/-} primary hepatocytes with/without 20 μM BAPTA pretreated for 1h and incubated at 45°C for 0,15, 30 or 60 min. Western blot analysis was performed with indicated antibodies.”

Q2: I am still somewhat concerned that the apparent somewhat variable self-degradation by CAPN3 in the *in vitro* cleavage experiments makes the comparison between the different PRC1 subunits difficult and it would be therefore helpful to show at least 2 replicates of each *in vitro* cleavage assay for all tested proteins.

A2: We understand your concerns and we have put the replicates of each figure below:

Figure 2 D and E:

Repeat of Figure 2 D and E:

Figure 4 B and C:

Repeat of Figure 4 B and C:

Figure EV2:

Repeat of Figure EV2:

Figure EV4:

Repeat of Figure EV4:

Q3: The advisor also mentioned that “partial hepatectomies are major operations and the quality of surgery/recovery will have an impact on their data” and we would therefore ask you to provide a detailed protocol of the surgery and a supplementary figure which shows the quality control measures that you have taken in controlling the quality and reproducibility of this experiment.”

A3: We agree that partial hepatectomy is a major surgical procedure, and its quality can significantly influence experimental outcomes. The surgeries were performed by a highly trained operator using a well-established and reproducible technique, which has been previously described in *Nature Protocol* (2008). To address the concerns raised, we have updated the Materials and Methods section to include a detailed surgical protocol. Additionally,

we have provided a schematic diagram of the partial hepatectomy procedure in Figure EV1D. In the surgery, the majority of the left lateral lobe and median lobe were resected, and their residues were excluded from subsequent analysis. For LBR, Western blot and RNA-seq analyses we used at least three individual mice per time point and per genotype, ensuring the reproducibility and consistency of the results. Specifically, we have also included LBR analysis for homozygous *RYBP*^{Y70A} mice (Fig. 5C).

These new data are now shown as Fig. EV1D and Fig. 5C and described in the text of the paper.

Page 12, paragraph 4:

“Partial hepatectomy

2/3 partial hepatectomy was performed based on the published protocol with modifications (Mitchell and Willenbring, 2008). Briefly, two-month-old male mice were anesthetized with pentobarbital sodium (50 mg/kg, i. p.). Following a midline laparotomy of 2 cm to expose the xiphoid process, the left lateral lobe was ligated near its base using 4-0 silk thread and subsequently resected. The median lobe was then ligated and resected, leaving a small portion of tissue (~3 mm from the suprahepatic vena cava) to maintain blood outflow from right and caudate lobes (Fig. EV1D). Subsequently, the peritoneum and skin were closed, animals were placed in a clean, warm recovery area until they regained consciousness. The procedure was carried out with care to minimize pain and distress. Animals were observed closely for the first 24 hours post-surgery and then twice daily for the next three days to monitor for any signs of complications, infection, or distress.”

Figure EV1D and legend:

Figure EV1. Generation of *Capn3* KO mice by CRISPR/Cas9 genome editing system

(D) Scheme of mouse liver anatomy and positioning of silk threads for knots (dash lines).

Figure 5C and legend:

Figure 5. RYBP^{Y70A} antagonizes the proteolysis of CAPN3 and inhibits the cell cycle reentry during liver regeneration

(C) The liver/body weight ratio (LBR) at different time points in 2 months old WT and RYBP^{Y70A} mice. All data represent mean \pm s.d., **, p < 0.01."

In summary, we have added more replicates to demonstrate the reproducibility of both the *in vitro* cleavage assays and the partial hepatectomy experiments. Additionally, our new data indicate that CAPN3 binds to RING1A under normal conditions in hepatocytes, and its activation is dependent on Ca⁺⁺ during stress response. We have also attached the updated manuscript (with revised texts in red) for your review. We believe that the revisions made have greatly enhanced the quality of our work. We hope that you will find the updated version to be a stronger candidate for publication in *The EMBO Journal*. If you are still interested in our work, we would be delighted to resubmit.

Thanks again for your time and consideration. Looking forward to your response.

Best,
Hui

Hui Shi, Ph.D.
ZJU100 Young Professor
MOE Key Laboratory of Biosystems Homeostasis & Protection
College of Animal Sciences
Zhejiang University
866 Yuhangtang Road
Hangzhou 310058, China
Hui_shi@zju.edu.cn

Dear Dr. Shi,

Thank you for submitting your manuscript for consideration by the EMBO Journal. It has now been seen by two referees whose comments are shown below. A third referee did not reply despite multiple reminders and we have therefore decided to proceed with the current two referees.

As you can see from the reports the referees appreciate the manuscript and think that the findings are interesting. However, both referees but especially referee #2 voice multiple major concerns regarding the technical quality of some of the experiments, as well as the lack of replicates. All together I find the referee reports productive and I would therefore like to invite you to submit a revised version of the manuscript, thoroughly addressing the comments of both reviewers. I should add that it is EMBO Journal policy to allow only a single round of revision, and acceptance of your manuscript will therefore depend on the completeness of your responses in this revised version.

Thank you for the opportunity to consider your work for publication. I look forward to your revision. If you have any questions do not hesitate to contact us. I am also happy to discuss the revisions via email or videoconferencing.

Yours sincerely,

Cornelius Schneider, PhD
Editor
The EMBO Journal
c.schneider@embojournal.org

We realize that it is difficult to revise to a specific deadline. In the interest of protecting the conceptual advance provided by the work, we recommend a revision within 3 months (19th Jun 2025). Please discuss the revision progress ahead of this time with the editor if you require more time to complete the revisions. Use the link below to submit your revision:

Referee #1:

Protease-mediated PRC1 dissociation promotes H2AK119ub remodeling during stress response by Wei Cui et. al is an excellent work demonstrating CAPN3-mediated PRC1 dissociation is responsible for rapidly rendering chromatin competent to stress signals. Especially, it's essential that authors revealed the contribution of direct degradation of PRC1 by other factors in regulating PRC-mediated repressive states. However, following issues could be discussed to further consolidate the authors' conclusion.

Major

1. Related to Figs. 1B and C, to make sure that change of the chromatin states proceeds the alteration of the cell cycle progression, authors should show the fractions of EdU positive hepatocytes prior to 16 hrs after resection.
2. It would be more persuasive if the effect on cell cycle progression is disclosed to be cause or consequence of destabilization of uH2A in Capn3 deleted cells. Is it possible to test occupancy of uH2A in Capn3 deleted cells while cell cycle progression is arrested?
3. I wonder if RYBPY70A preserves the properties of PRC1 components. Therefore, the authors should show if RYBPY70A forms complexes with RING1B, KDM2B, or like.

Minor

In Fig. 1E, it could be favorable if the areas where uH2A occupancy is not changed are shown.

Referee #2:

CAPN3 is a calcium-activated cysteine protease previously implicated in zebrafish liver development and regeneration. In this study, the authors investigate how CAPN3 loss affects Polycomb gene regulation under stress conditions induced by partial hepatectomy or heat shock. They demonstrate that CAPN3 mediates the proteolysis of multiple PRC1 subunits, which may contribute to a reduction in the PRC1 histone mark, H2AK119ub. Focusing on the PRC1 subunit RYBP, they show that CAPN3-dependent proteolysis is enhanced in the presence of the core PRC1 component RING1A and identify a point mutant of RYBP that resists degradation. Their findings suggest that CAPN3-mediated proteolysis of PRC1 subunits alleviates PRC1 repression, enabling target genes to become active during cell proliferation and stress responses.

This paper proposes a novel mechanism by which CAPN3 regulates the Polycomb mark H2AK119ub, which is likely to be of broad interest. However, some of the experimental results in support of the conclusions are not fully convincing and some additional work is required before I can recommend publication. Specific points follow below.

Major points

1. Experimental replicates. Throughout the study, information about replicates is missing. This is particularly problematic for "big data" experiments such as the ChIP-seq in Fig. 1 and the mass spec in Fig. 2. If replicates were performed this should be specified and a statistical analysis included. If not, given the quantitative differences claimed, the experiments must be repeated at least once.
2. ChIP-seq. Given the quantitative nature of the effect claimed the authors should include replicates and ideally also spike-ins for normalization. In addition it would be helpful to see a WB that confirms the differences in H2AK119ub.
3. RNA-seq. The RNA-seq analysis seems cherry picked, only showing two examples in the genome browser and a few genes in the heatmap. A more comprehensive analysis of the RNA-seq and comparison with the ChIP-seq is recommended.
4. Mass spec analysis. In Fig. 2B, the number of unique peptides is not an appropriate measure. Please provide protein

abundance by iBAQ or similar.

5. Fig. 2C: why is the truncated form of RYBP preferentially IP'ed? This should be repeated.

6. Fig. 3C. it is hard to come to any conclusion with this figure because the levels of the overexpressed proteins are very different. It is particularly puzzling that PCGF6 cannot be seen in the input or in the IP and yet it co-purified with CAPN3. Also the authors should comment on the band at ~50 kDa in all IPs.

7. Capn3 KO and RYBP mutant. To corroborate the proposed model the authors could show that cells carrying the Rybp mutation do not show any additional change in the Polycomb pathway when Capn3 is missing.

8. Capn3 WT and mutant overexpression. Why is the mobility of this mutant affected so dramatically in every WB even though it is just a single substitution? Why are the levels of Capn3 WT so much lower than the C129S mutant in all experiments?

9. Stress and hepatectomy. The link between heat shock (Fig. 6) and liver regeneration (Fig. 1, 2, 5) needs to be better explained.

10. Fig 6D: The model suggests that core PRC1 dissociates from chromatin, but there is no direct evidence supporting this. Fig 6C only shows a decrease in CBX8 and H2AK119ub, neither of which are core PRC1 components. Both chromatin and nuclear fractionation should be performed to assess core PRC1 subunits, such as RING1A/B.

Minor points

- Fig. 1B, 5G: the y axis shows "fraction" but the legend says "percentage".

- Fig. 1H: the alluvial plot is very hard to read and it is unclear what is the conclusion. Why is 16 h Capn3 presented before 0 h? Why is there overall more state #4 at 16h than at 0h in Capn3? This is not consistent with the dynamics of single marks shown in Fig. 1E.

- Fig. 1J: mistaken call in the text?

- Fig. 2A: I do not see a "drastic decrease". This should be repeated and quantified.

- Fig. 2H: it seems that RYBP is decreased at 16h even in absence of CAPN3

- Fig. EV3 and EV5 seem to be missing

- Fig. 5D-E: not called in the text

- A potential role for the H2AK119ub deubiquitinase PR-DUB should be discussed in the text.

- CAPN3 has low expression in the liver compared to tissue like muscle. Does this protein play a similar role in muscle growth or is this unique to hepatocytes and regeneration?

- For IPs the % of input should be indicated in the figures.

- Both the introduction and discussion mention "response to mitotic signals" being a key investigation in the paper, but only cell proliferation in general was discussed.

- The text should address the distinction between canonical and non-canonical PRC1 in regards to H2AK119ub deposition.

Response to Reviewers

We were pleased to receive a favorable review of our manuscript (EMBOJ-2024-118128R1-Q), entitled "Protease-mediated PRC1 dissociation promotes H2AK119ub remodeling during stress response". We now address each of the reviewers' comments in detail, indicating the pages of the manuscript where corresponding changes have been made. We have performed a substantial number of new experiments and have revised our interpretations in the manuscript as requested by the reviewers. We hope that with these modifications, our manuscript will be acceptable for publication in The EMBO Journal.

Response to Reviewer #1:

Thank you for expert review of our manuscript. We have now clarified each of the issues raised and performed new experiments that have been included in the manuscript. We believe that this has significantly improved the manuscript. A point-by-point response is included below:

Reviewer 1 comment: *A Protease-mediated PRC1 dissociation promotes H2AK119ub remodeling during stress response by Wei Cui et. al is an excellent work demonstrating CAPN3-mediated PRC1 dissociation is responsible for rapidly rendering chromatin competent to stress signals. Especially, it's essential that authors revealed the contribution of direct degradation of PRC1 by other factors in regulating PRC-mediated repressive states. However, following issues could be discussed to further consolidate the authors' conclusion.*

Major:

1. *Related to Figs. 1B and C, to make sure that change of the chromatin states proceeds the alteration of the cell cycle progression, authors should show the fractions of EdU positive hepatocytes prior to 16 hrs after resection.*

Authors' response: We sincerely appreciate your insightful suggestion. We have performed additional EdU pulse-labeling coupled with Hnf4 α immunofluorescence staining for both wild-type and *Capn3*^{-/-} mice at 0 h and 16 h post-PHx. We did not detect any EdU-positive hepatocytes at this time point in either genotype, indicating that the resumption of hepatocyte proliferation had not yet commenced by 16 h post-PHx.

These new data are now shown as Fig. 1B and Fig. EV2B and described in the text of the paper.

Page 5, paragraph 1:

"Compared to wild-type mice, the fractions of EdU-positive hepatocytes in *Capn3*^{-/-} mice were significantly reduced at both 24 and 38 hours post-PHx (0.2% vs 6.9%,

$p < 0.001$ and 3.3% vs 30.8%, $p = 0.015$, respectively), while neither genotype exhibited detectable EdU-positive hepatocytes at 0 h and 16 h post-PHx (Figs. 1B and EV 2B)."

Fig. 1B and legend:

Figure 1. Depletion of CAPN3 inhibits H2AK119ub remodeling and delays the resumption of proliferation in hepatocytes after PHx

(B) Quantification of the fractions of EdU⁺ hepatocytes at 0, 16, 24, and 38 h post-PHx. Data represent mean \pm s.d., *, $p < 0.1$; ***, $p < 0.001$, $n = 3$.

Fig. EV2B and legend:

Figure EV2. Depletion of CAPN3 delays hepatocyte proliferation and inhibits H2AK119ub remodeling after PHx

(B) Immunostaining of EdU and Hnf4a (a hepatocyte marker) at 0 h and 16 h post-PHx. Scale bars, 40 μ m.

2. *It would be more persuasive if the effect on cell cycle progression is disclosed to be cause or consequence of destabilization of uH2A in Capn3 deleted cells. Is it possible to test occupancy of uH2A in Capn3 deleted cells while cell cycle progression is arrested?*

Authors' response: Thank you for raising this meaningful point. We totally agree that it would be more persuasive if we could figure out that whether the delay of cell cycle progression is to be cause or consequence of the inhibition of H2AK119ub deubiquitylation in Capn3 deleted cells. In adult mammals, hepatocytes are generally quiescent, with cell cycle progression arrested. Thanks to your *major suggestion #1*, we have verified that hepatocytes at 16 h post-PHx have not yet commenced proliferation, indicating that uH2A deubiquitylation occurs prior to the onset of proliferation. Accordingly, we have rephrased the first subtitle in the "Results" section of our revised manuscript as follows:

Page 4, paragraph 5:

"Depletion of CAPN3 inhibits H2AK119ub remodeling and further delays the resumption of proliferation in hepatocytes after PHx"

3. *I wonder if RYBP^{Y70A} preserves the properties of PRC1 components. Therefore, the authors should show if RYBP^{Y70A} forms complexes with RING1B, KDM2B, or like.*

Authors' response: Thank you for your valuable suggestion. We have performed new experiments to address this question. To investigate whether RYBP^{Y70A} retains the ability to form complexes with other PRC1 components, we overexpressed either Flag-RYBP or Flag-RYBP^{Y70A} in 293T cells and performed immunoprecipitation with anti-Flag beads. Western blotting analysis revealed that both wild-type and mutant RYBP were able to form complexes with RING1A/B and PCGF3/5, indicating that RYBP^{Y70A} preserves the properties of PRC1 components. However, KDM2B were not detectable in immunoprecipitation with either wide-type or mutant RYBP.

These new data are now shown as Fig. 4E and described in the text of the paper.

Page 9, paragraph 1:

"To further assess whether the RYBP^{Y70A} mutation preserves the properties of PRC1 components, we overexpressed either Flag-RYBP or Flag-RYBP^{Y70A} in 293T cells and performed immunoprecipitation using anti-Flag beads. Subsequent western blot analysis revealed that both wild-type and mutant RYBP co-immunoprecipitated with RING1A/B and PCGF3/5, indicating that RYBP^{Y70A} maintains its ability to form PRC1 complexes (Fig. 4E)."

Fig. 4E and legend:

Figure 4. Tyr⁷⁰ of RYBP was the key amino acid for its proteolysis by CAPN3
(E) Western blot analysis on anti-Flag immunoprecipitated from nuclear extracts of 293T cells transfected with 500 ng *Flag-RYBP-pCS2+*, 500ng *Flag-RYBP^{Y70A}-pCS2+* or 500ng *pCS2+*. Cell lysates were harvested after 48 h. 'Input' contains ~10% of the input cell lysate used for IP. Antibodies used are indicated.

Minor:

In Fig. 1E, it could be favorable if the areas where uH2A occupancy is not changed are shown.

Authors' response: Thank you for this suggestion. We have added a new IGV track example highlighting regions where uH2A occupancy is either unchanged or altered.

These new data are now shown as Fig. 1K.

Fig. 1K and legend:

Figure 1. Depletion of CAPN3 inhibits H2AK119ub remodeling and delays the resumption of proliferation in hepatocytes after PHx

(K) Genome browser snapshots (left) of H2AK119ub, H3K27me3 and H3K27ac tracks at the *Adgrg1* loci in WT and *Capn3*^{-/-} hepatocytes at 0 h or 16 h post-PHx. The expression levels (right) of *Adgrg1* in TPM at 0 h or 16 h post-PHx are summarized in bar charts. The yellow shaded areas highlight #42_44 regions, while the grey shaded area marks a #44_44 region with H2AK119ub signals unchanged across the four conditions.

Response to Reviewer #2:

Thank you for expert review of our manuscript. We have now clarified each of the issues raised and performed new experiments that have been included in the manuscript. We believe that this has significantly improved the manuscript. A point-by-point response is included below:

Reviewer 2 comment: *CAPN3 is a calcium-activated cysteine protease previously implicated in zebrafish liver development and regeneration. In this study, the authors investigate how CAPN3 loss affects Polycomb gene regulation under stress conditions induced by partial hepatectomy or heat shock. They demonstrate that CAPN3 mediates the proteolysis of multiple PRC1 subunits, which may contribute to a reduction in the PRC1 histone mark, H2AK119ub. Focusing on the PRC1 subunit RYBP, they show that CAPN3-dependent proteolysis is enhanced in the presence of the core PRC1 component RING1A and identify a point mutant of RYBP that resists degradation. Their findings suggest that CAPN3-mediated proteolysis of PRC1 subunits alleviates PRC1 repression, enabling target genes to become active during cell proliferation and stress responses.*

This paper proposes a novel mechanism by which CAPN3 regulates the Polycomb mark H2AK119ub, which is likely to be of broad interest. However, some of the experimental results in support of the conclusions are not fully convincing and some additional work is required before I can recommend publication. Specific points follow below.

Major points:

1. Experimental replicates. *Throughout the study, information about replicates is missing. This is particularly problematic for "big data" experiments such as the ChIP-seq in Fig. 1 and the mass spec in Fig. 2. If replicates were performed this should be specified and a statistical analysis included. If not, given the quantitative differences claimed, the experiments must be repeated at least once.*

Authors' response: We appreciate the reviewer's insightful comment regarding experimental replicates.

The ChIP-seq experiments depicted in Fig. 1 were conducted using hepatocytes pooled from three independent biological replicates to ensure the acquisition of representative and high-quality signals. We have now performed two independent CUT&Tag assays for H2AK119ub using two biological samples per genotype to verify reproducibility. The overall trend of H2AK119ub signals in the new replicates is consistent with the previous ChIP-seq results, strongly supporting our initial findings.

These new data are now shown as Fig. EV2D and described in the text of the paper.

Page 5, paragraph 3:

"The differential dynamics of H2AK119ub between wild-type and *Capn3*^{-/-} hepatocytes

at 16 h post-PHx were further confirmed by CUT&Tag analysis and western blot analysis of chromatin fractions (Fig. EV 2D, E).”

Fig. EV2D and legend:

Figure EV2. Depletion of CAPN3 delays hepatocyte proliferation and inhibits H2AK119ub remodeling after PHx

(D) Two independent biological replicates of CUT&Tag profiles and heat maps of H2AK119ub in WT and *Capn3*^{-/-} hepatocytes at 0 h and 16 h post-PHx at called peaks. Color intensity for each strand represents counts per million (CPM).

Regarding the mass spec analysis of CAPN3 overexpression, we have conducted three additional biological replicates and performed statistical analysis on the new data. Three non-core components of PRC1 complexes showed significantly reduced intensities.

These updated results are now shown as Fig. 2B and described in the text of the paper.

Page 7, paragraph 3:

“To further verify our hypothesis, we generated another HepG2 cell line with doxycycline-inducible HA-CAPN3^{C129S} expression (Tet-HA-CAPN3^{C129S}). CAPN3^{C129S} is an enzymatic inactive mutant of CAPN3 whose catalytic core residue Cys¹²⁹ was mutated to serine, serving as a negative control (Sorimachi et al., 1993; Tao et al., 2013). After 48 h of doxycycline treatment, Ctrl, Tet-HA-CAPN3 and Tet-HA-CAPN3^{C129S} cells were harvested, and the nuclear protein fraction was subsequently extracted. The nuclear extracts were affinity purified by RING1A antibody and then analyzed by mass spectrometry (MS). The MS analysis revealed that overexpression of CAPN3 resulted in a significant reduction in several non-core PRC1 components, including CBX8, RYBP and YAF2 (Fig. 2B).”

Fig. 2B and legend:

Figure 2. CAPN3 regulates H2AK119ub remodeling by proteolysis of PRC1 non-core components during liver regeneration.

(B) The Mass spectrometry (MS) analysis of RING1A immunoprecipitation (IP) from nuclear protein extracts of Ctrl, Tet-HA-CAPN3 and Tet-HA-CAPN3^{C129S} HepG2 cells treated with 1 μ g/mL doxycycline for 48 h. Each sample contains about 2×10^7 cells. The numbers and color intensity indicate the mean of log₂-transformed Label-Free Quantification (LFQ) intensity values detected in the MS analysis. n = 3, *, p < 0.05; **, p < 0.01.

2. ChIP-seq. Given the quantitative nature of the effect claimed the authors should include replicates and ideally also spike-ins for normalization. In addition it would be helpful to see a WB that confirms the differences in H2AK119ub.

Authors' response: We appreciated your valuable suggestions. As previously addressed in our response to your first major inquiry, the ChIP-seq experiments depicted in Fig. 1 were performed using hepatocytes pooled from three biological replicates. In alignment with your recommendation, we conducted two independent CUT&Tag assays for H2AK119ub, using two biological samples per genotype to ensure reproducibility. These assays were executed with the NovoNGS® CUT&Tag 4.0 High-Sensitivity kit (Novoprotein, N259-YH01) and incorporated spike-ins. However, we found that the inclusion of spike-ins is impractical for hepatocytes due to the high ratio of polyploid nuclear content present in hepatocytes. The application of spike-ins under these conditions would result in inaccurate normalization and potentially compromise the data integrity. Consequently, we adopted counts per million (CPM) to normalize the sequencing depth across all samples. Nonetheless, the CUT&Tag analysis of H2AK119ub with spike-ins is also presented below, showing a more pronounced trend of change between wild-type and *Capn3*^{-/-} compared to the data in Fig. EV2D, which were normalized by CPM:

Figure for reviewer: CUT&Tag profiles and heat maps of H2AK119ub in WT and *Capn3*^{-/-} hepatocytes at 0 h and 16 h post-PHx at called peaks. Color intensity for each strand represents normalized reads based on the spike-in.

We strongly agree that a chromatin WB would be an excellent way to assess the differences in H2AK119ub. That's really a great suggestion.

The corresponding new data are now shown in Fig EV2E.

Fig. EV2E and legend:

Figure EV2. Depletion of CAPN3 delays hepatocyte proliferation and inhibits H2AK119ub remodeling after PHx

(E) Chromatin fractions were isolated from the livers of three independent WT and *Capn3*^{-/-} mouse. Western blot analysis was performed with indicated antibodies.

3. RNA-seq. The RNA-seq analysis seems cherry picked, only showing two examples in the genome browser and a few genes in the heatmap. A more comprehensive analysis of the RNA-seq and comparison with the ChIP-seq is recommended.

Authors' response: Thank you for your suggestion. We have performed more comprehensive analysis of the RNA-seq and comparison with the ChIP-seq.

These new data are now shown as Fig. 1 H-L and EV2F-H and described in the text of the paper.

Page 6, paragraphs 1 and 2:

“To evaluate whether such disruption of chromatin remodeling would further affect downstream gene expression at 16 h or later, we performed RNA-seq analysis on wild-type and *Capn3*^{-/-} hepatocytes at 0, 16, 24, and 36 h post-PHx (Fig. 1C). By integrating the ChIP-seq and transcriptomic data at 0 h and 16 h, we identified 44 genes proximal to these disrupted loci whose induction at 16 h was significantly attenuated in *Capn3*^{-/-} hepatocytes compared to wild-type hepatocytes, suggesting their full activation may relate to the chromatin remodeling (Figs. 1I and EV2F). Gene Ontology (GO) analysis of the 44 genes revealed significant enrichment for molecular functions (MF) related to extracellular matrix interactions (e.g. collagen, glycosaminoglycan, and laminin binding, extracellular matrix structural constituent), growth factor signaling modulation (platelet-derived growth factor and growth factor binding), and key metabolic and regulatory functions (heme and tetrapyrrole binding, serine-type endopeptidase inhibitor activity), suggesting a role in preparing for signal responses (Fig. 1J and K).

To further assess the later impact of chromatin remodeling disruption at 16 h post-PHx on downstream genes, we subjected genes annotated to #41_44, #42_44, and #43_44 (detected at > 0.1 TPM in at least one conditions of the 16 h, 24 h, and 36 h) to model-based clustering using Mclust into 16 clusters (Fig. EV2G and H) (Scrucca et al. 2016). Cluster7 and 16 showed increased gene expression at 24 h and 36 h in wild-type hepatocytes, while gene expression levels remained stable across 16, 24, and 36 h in *Capn3*^{-/-} hepatocytes (Fig. EV2 H). Interestingly, Cluster4 exhibited delayed expression in *Capn3*^{-/-} hepatocytes, peaking at 36 h instead of 24 h as seen in wild-type hepatocytes, including transcription related genes (e.g. *Etv5*, *Tcf20*, *Phtf2*), mRNA maturation related genes (e.g. *Alkbh5*, *Thoc3*, *Thrc6c*), protein synthesis and transport related genes (e.g. *Utp6*, *Ago2*, *Tars*), and a crucial cell cycle regulator, *Cdc5l* (Figs. EV2H and 1L). These findings suggest that the disruption of chromatin remodeling in *Capn3*^{-/-} hepatocytes at 16 h post-PHx continues to impair the subsequent induction of downstream genes during later stages of liver regeneration.”

Fig. 1 H-L and legends:

Figure 1. Depletion of CAPN3 inhibits H2AK119ub remodeling and delays proliferation in hepatocytes after PHX

(H) Distribution of different subset of regions overall genome obtained by ChIP-seq experiments. The four numerical codes for genomic regions denote the chromatin state classifications across the four conditions: WT-PHX-0 h, WT-PHX-16 h, *Capn3*^{-/-}-PHX-0 h, and *Capn3*^{-/-}-PHX-16 h. For instance, #41_44 represents “4→1 in WT vs.4→4 in *Capn3*^{-/-}”, with the other codes following the same convention.

(I) Venn diagrams showing the extent of overlap for the genes near #41_44, #42_44 and #43_44 loci, genes upregulated in WT hepatocytes at 16 h post-PHX vs. 0h, and genes downregulated in *Capn3*^{-/-} hepatocytes at 16 h post-PHX relative to WT. The intersection of these three gene sets is highlighted by the red dotted-line frame.

(J) GO term enrichment analysis of the 44 genes in (I).

(K) Genome browser snapshots (left) of H2AK119ub, H3K27me3 and H3K27ac tracks at the *Adgrg1* loci in WT and *Capn3*^{-/-} hepatocytes at 0 h or 16 h post-PHX. The expression levels (right) of *Adgrg1* in TPM at 0 h or 16 h post-PHX are summarized in bar charts. The yellow shaded areas highlight #42_44 regions, while the grey shaded area marks a #44_44 region with H2AK119ub signals unchanged across the four conditions.

(L) Heatmap showing the 87 delayed up-regulated genes in the *Capn3*^{-/-} in Cluster4 defined by Mclust in Figure EV2 (G).

Fig. EV2F-H and legend:

Figure EV2. Depletion of CAPN3 delays hepatocyte proliferation and inhibits H2AK119ub remodeling after PHx

(F) Genome browser snapshots (left) of H2AK119ub, H3K27me3 and H3K27ac tracks at the *Cd63* loci in WT and *Capn3*^{-/-} hepatocytes at 0 h or 16 h post-PHx. The expression levels (right) of *Cd63* in TPM at 0 h or 16 h post-PHx are summarized in bar charts. The shaded areas highlight the #41_44 (blue) and #42_44 (yellow) regions.

(G) Bayesian information content (BIC) was computed to assess 2–30 component models to get the optimal cluster number (indicated by orange arrow).

(H) Covariance clustering of #41_44, #42_44, and #43_44-related genes (TPM>0.1 in ≥ 1 condition, n=1232) across 16, 24, and 36 h PHx in WT and *Capn3*^{-/-} hepatocytes into 16 clusters. Gene counts for each cluster are shown at the bottom right of each corresponding panel.

4. Mass spec analysis. In Fig. 2B, the number of unique peptides is not an appropriate measure. Please provide protein abundance by iBAQ or similar.

Authors' response: Thank you for your suggestion. In response, we have analyzed our new mass spec data by applying a log₂ transformation to the Label-Free Quantification (LFQ) intensity values. The resulting heatmap illustrates the mean log₂ LFQ intensity of PRC1 components across replicates for each cell line, with significant changes relative to the control (Ctrl) denoted by asterisks.

These updated results are now shown as Fig. 2B and described in the text of the paper.

Page 7, paragraph 3:

“To further verify our hypothesis, we generated another HepG2 cell line with doxycycline-inducible HA-CAPN3^{C129S} expression (Tet-HA-CAPN3^{C129S}), where CAPN3^{C129S} is an enzymatic inactive mutant of CAPN3 whose catalytic core residue Cys¹²⁹ mutated to serine, serving as a negative control (Sorimachi et al., 1993; Tao et al., 2013). After 48 h of doxycycline treatment, Ctrl, Tet-HA-CAPN3 and Tet-HA-CAPN3^{C129S} cells were harvested, and the nuclear protein fraction was subsequently extracted. The nuclear extracts were affinity purified by RING1A antibody and then analyzed by mass spectrometry (MS). The MS analysis revealed that overexpression of CAPN3 resulted in a significant reduction in several non-core PRC1 components, including CBX8, RYBP and YAF2 (Fig. 2B).”

5. Fig. 2C: why is the truncated form of RYBP preferentially IP'ed? This should be repeated.

Authors' response: We apologize for any lack of clarity in the original manuscript. We hypothesize that RYBP within the complex is preferentially cleaved by CAPN3. Therefore, the ratio of the IP'ed truncated form to the full-length of RYBP is much higher than in the input lysate. Multiple experimental repeats confirmed this observation, and representative data from two independent repeats are shown below:

Figure for reviewer: Western blot analysis of RING1A IP from nuclear protein extracts of Ctrl, Tet-HA-CAPN3 and Tet-HA-CAPN3^{C129S} HepG2 cells treated with 1 µg/mL doxycycline for 48 h. Each sample contains about 2X10⁷ cells. ‘Input’ contains ~10% of the input cell lysate used for IP. Antibodies used are indicated.

*The western blot transfer of the input was uneven on the left side of the 55 kDa region, resulting in the abnormal appearance of the Ring1A and LaminB1 bands in the Ctrl input.

We also appreciate you for raising this important point. To eliminate the possibility that the observed result is due to preferential immunoprecipitation of the truncated form of RYBP, we constructed a Flag-tagged truncated RYBP variant (Flag-RYBP^{Δ1-70}) and conducted additional co-IP experiments to compare the binding affinities of the full-length RYBP and RYBP^{Δ1-70} with RING1A. Subsequent western blot analysis revealed comparable binding affinities of full-length RYBP and RYBP^{Δ1-70} for RING1A,

suggesting the elevated ratio of truncated RYBP in IP complexes is likely due to the preferential cleavage of complex-associated RYBP by CAPN3. We sincerely appreciate your insightful comment and believe this additional evidence strengthens our conclusion.

These new data are now shown below:

Figure for reviewer: Western blot analysis on anti-Flag immunoprecipitated from nuclear extracts of 293T cells co-transfected with *HA-RING1A-pCS2+* and *pCS2+*, *Flag-RYBP-pCS2+* or *Flag-RYBP^{Δ1-70}-pCS2+*. 'Input' contains ~10% of the input cell lysate used for IP. Antibodies used are indicated. Red asterisk indicates a non-specific band.

6. Fig. 3C. it is hard to come to any conclusion with this figure because the levels of the overexpressed proteins are very different. It is particularly puzzling that PCGF6 cannot be seen in the input or in the IP and yet it co-purified with CAPN3. Also the authors should comment on the band at ~50 kDa in all IPs.

Authors' response: Thank you for pointing this out. In the initial figure, the band at ~50 kDa in all IPs corresponds to the IgG heavy chain, which co-migrates with and obscures the detection of RING1A, PCGF2, and particularly PCGF6, all of which migrate near 50 kDa. To address this issue, we have now utilized anti-Flag Nanobody Magarose Beads (AlpaLifeBio, Cat. No. KTSM1338), which effectively eliminate the interfering IgG bands in subsequent western blot analysis of IPs. The results indicate that, compared to Flag-tagged CBX2, both Flag-tagged RING1A and PCGF2 exhibited a higher affinity for Myc-CAPN3, as evidenced by the greater amount of Myc-CAPN3 pulled down when present at comparable levels. Notably, PCGF6 demonstrated a robust interaction by pulling down more Myc-CAPN3 than both CBX2 and RYBP, even with markedly less input protein.

These new data are shown below:

Figure for reviewer: Western blot analysis on anti-Flag immunoprecipitated from cell lysates using anti-Myc and anti-Flag antibodies. 293T cells were co-transfected with *Myc-CAPN3^{C129S}-pCS2+* and *Flag-RING1A-pCS2+*, *Flag-RING1B-pCS2+*, *Flag-PCGF2-pCS2+*, *Flag-PCGF6-pCS2+*, *Flag-CBX2-pCS2+*, *Flag-RYBP-pCS2+*, or *pCS2+*. Cell lysates were harvested after 48 h. 'Input' contains ~10% of the input cell lysate used for IP.

It remains challenging to adjust all overexpressed PRC1 components to uniform levels within a single IP experiment. To enhance the rigor of our study, we have revised our manuscript to emphasize the higher affinity of RING1A and its role in facilitating the proteolysis of PRC1 non-core components by CAPN3.

These new data are now shown as Fig. 3C and described in the text of the paper.

Page 8, paragraph 3:

"To test whether this effect is due to a higher affinity of RING1A for CAPN3, facilitating its recruitment, we overexpressed Myc-tagged CAPN3^{C129S} together with Flag-tagged RING1A and several PRC1 non-core components and performed co-immunoprecipitation analyses. The results showed that CAPN3^{C129S} had a higher binding affinity for the PRC1 core component RING1A compared to non-core components like RYBP, CBX2 and CBX8. This suggests that the PRC1 core component RING1A may facilitate the recruitment of CAPN3 for the proteolysis of PRC1 non-core components (Fig. 3C)."

Fig. 3C and legend:

Figure 3. PRC1 core component RING1A helps to recruit CAPN3 to proteolyze non-core components

(C) Western blot analysis on anti-Flag immunoprecipitated from cell lysates using anti-Myc and anti-Flag antibodies. 293T cells were co-transfected with 500ng *Myc-CAPN3^{C129S}*-pCS2+ and 1ug pCS2+, 500ng *Flag-RING1A*-pCS2+, 500ng *Flag-RYBP*-pCS2+, 500ng *Flag-CBX2*-pCS2+, 500ng *Flag-CBX8*-pCS2+ or 1ug *Flag-RING1A*-pCS2+. Cell lysates were harvested after 48h. 'Input' contains ~10% of the input cell lysate used for IP. This experiment was replicated at least twice.

7. Capn3 KO and RYBP mutant. To corroborate the proposed model the authors could show that cells carrying the Rybp mutation do not show any additional change in the Polycomb pathway when Capn3 is missing.

Authors' response: We apologize for the lack of clarity in the original manuscript. While we have demonstrated that CAPN3-mediated proteolysis of RYBP is critical for timely liver regeneration, it is important to note that RYBP is not the sole substrate of CAPN3. We have not excluded the potential roles of other non-core components of PRC1 in this process. In addition, we have performed more experiments using the RYBP mutant to further validate our model.

These new data are now shown as Fig. 5C and EV5E and described in the text of the paper.

Page 9, paragraph 3:

"To evaluate whether liver regeneration was affected in *RYBP^{Y70A}* knock-in mice, we conducted 2/3 PHx on both wild-type and homozygous *RYBP^{Y70A}* knock-in mice. LBR were measured at 0, 1, 3 and 5 days post-PHx. The results showed a significant delay in LBR recovery during liver regeneration in homozygous *RYBP^{Y70A}* knock-in mice (Fig. 5C)."

Fig. 5C and legend:

Figure 5. RYBP^{Y70A} knock-in mice exhibit delayed liver regeneration after PHx
(C) The liver/body weight ratio (LBR) at different time points before and post-PHx in 2 months old WT and RYBP^{Y70A} mice. All data represent mean \pm s.d., **, $p < 0.01$.

Page 10, paragraph 2:

“A similar delayed PRC1 dissociation from chromatin and H2AK119ub reduction after heat shock was observed in primary hepatocytes derived from homozygous RYBP^{Y70A} knock-in mice, further supporting the role of the CAPN3-PRC1 pathway in H2AK119ub reduction during stress response (Fig. EV5E).”

Fig. EV5E and legend:

Figure EV5. CAPN3 facilitates nucleolus entry of RYBP and H2AK119ub elimination during stress response

(C) Chromatin fractions were isolated from WT, RYBP^{Y70A} homozygous and Capn3^{-/-} primary hepatocytes incubated at 45°C for 0, 15, 30, or 60 min. Western blot analysis was performed with indicated antibodies. This experiment was replicated at least twice.

8. Capn3 WT and mutant overexpression. Why is the mobility of this mutant affected so dramatically in every WB even though it is just a single substitution? Why are the levels of Capn3 WT so much lower than the C129S mutant in all experiments?

Authors' response: We apologize for the lack of clarity in the original manuscript regarding the autolytic activity of CAPN3. CAPN3 exhibits strong autolytic activity, which is abolished by mutating its catalytic core residue C129 to serine (C129S). Consequently, the levels of overexpressed CAPN3 WT are much lower than the C129S mutant (One et al., 2016).

This property has now been described in the revised manuscript, which is reprinted below:

Page 4, paragraph 1:

“It functions as a calcium-activated protease and undergoes extremely rapid and exhaustive autolytic activity, which is abolished when its catalytic core residue is mutated (Ono et al., 2016; Sorimachi et al., 1989).”

9. Stress and hepatectomy. The link between heat shock (Fig. 6) and liver regeneration (Fig. 1, 2, 5) needs to be better explained.

Authors' response: We appreciate this insightful suggestion. CAPN3 functions as a calcium-activated protease. Considering that both PHx and heat shock elevate Ca²⁺ signaling (Oliva et al., 2018), we hypothesize that these stressors activate CAPN3 via calcium flux. We conducted *in vitro* cleavage assays for RYBP in the presence and absence of free calcium ions. The results demonstrated that RYBP was cleaved by CAPN3 exclusively in the presence of free calcium ions, confirming that this reaction is dependent on calcium ions. Given the extensive influence of Ca²⁺ signaling on physiological processes, we opted to test this hypothesis using the *in vitro* heat shock model. Hepatocytes, pretreated with or without the calcium chelator BAPTA, were incubated at 45 °C for 0, 15, 30, or 60 minutes. Subsequent western blot analysis of chromatin revealed BAPTA treatment delayed chromatin clearance of H2AK119ub and CBX8 in wild-type hepatocytes during heat shock. This effect was not observed in *Capn3*^{-/-} samples, suggesting that stress-induced calcium fluctuation may play a crucial role in activating the CAPN3-PRC1 regulatory pathway.

These new data are now shown as Fig. 6E and F and described in the text of the paper.

Page 10, paragraph 1:

“CAPN3 is a calcium-dependent cysteine protease. We then performed the *in vitro* cleavage assay for RYBP in both the presence and absence of free calcium ions. The results indicated that RYBP was cleaved by CAPN3 only in the presence of free calcium ions, demonstrating that this reaction is calcium ion-dependent (Fig. 6E). Thus, we hypothesized that CAPN3 binds to PRC1 complexes prior to a response to stress signals and is likely activated by free calcium ions induced by cellular stresses. To test this hypothesis, we pre-treated hepatocytes with the calcium chelator BAPTA and incubated them at 45 °C for 0, 15, 30, or 60 min. Chromatin was extracted and western blot analysis was performed. The results indicated that BAPTA treatment delayed the elimination of both H2AK119ub and CBX8 from chromatin in wild-type hepatocytes during heat shock, but not in *Capn3*^{-/-} samples, suggesting that stress induced calcium fluctuation might play a key role in activating the CAPN3-PRC1 regulatory pathway (Fig. 6F).”

Page 12, paragraph 4:

“In addition, we found that chelating calcium attenuated the effect of heat shock on reducing the H2AK119ub level. Considering that PHx can also elevate Ca²⁺ signaling (Oliva et al., 2018) and that CAPN3 complexes with the PRC1, we propose that

calcium acts as an activator of the CAPN3-PRC1 pathway in response to stress signals such as PHx and heat shock. Overall, our study demonstrated the role of CAPN3 in facilitating the dissociation of PRC1 from chromatin during H2AK119ub remodeling, unveiling a novel protease-mediated epigenetic regulation system in rapid response to stress.”

Fig. 6E, F and legend:

Figure 6. Proteolysis of PRC1 non-core components by CAPN3 disrupts the maintenance of genomic H2AK119ub by PRC1 occupation

(E) Western blot analysis of *in vitro* CAPN3 proteolysis assays of human RYBP. Reaction mixture (containing 2 mM EDTA, chelated endogenous Ca²⁺) was incubated at 37°C for 10 min with or without 2.5 mM Ca²⁺ as indicated. This experiment was replicated at least twice.

(F) Chromatin fractions were isolated from WT and *Capn3*^{-/-} primary hepatocytes with/without 20 μM BAPTA pretreated for 1h and incubated at 45°C for 0, 15, 30, or 60 min. Western blot analysis was performed with indicated antibodies. This experiment was replicated at least twice.

10. Fig 6D: The model suggests that core PRC1 dissociates from chromatin, but there is no direct evidence supporting this. Fig 6C only shows a decrease in CBX8 and H2AK119ub, neither of which are core PRC1 components. Both chromatin and nuclear fractionation should be performed to assess core PRC1 subunits, such as RING1A/B.

Authors’ response: Thank you for this valuable suggestion. In response, we conducted additional heat shock experiments and concurrently collected total cell extracts, nuclear extracts and chromatin fractions from both wild-type and *Capn3*^{-/-} hepatocytes across identical time points. The findings indicated that the reduction in RING1A levels after heat shock was specifically detected in the chromatin fraction of wild-type cells, with this decrease being delayed in *Capn3*^{-/-} cells.

These new data are now shown as Fig. 6D and described in the text of the paper.

Page 12, paragraph 4:

“Moreover, the levels of chromatin-bound PRC1 core component RING1A also rapidly declined after heat shock in wild-type cells, while the abundance of RING1A in the total cell extracts and nuclear extracts remained unchanged (Fig. 6D). The reduction of chromatin-bound RING1A and H2AK119ub were both inhibited in *Capn3*^{-/-} cells, indicating that CAPN3 promotes the dissociation of PRC1 from chromatin and the subsequent reduction of H2AK119ub following heat shock (Fig. 6C and D).”

Fig. 6D and legend:

Figure 6. Proteolysis of PRC1 non-core components by CAPN3 disrupts the maintenance of genomic H2AK119ub by PRC1 occupation

(D) Total cell extracts, nuclear extracts and Chromatin fractions were isolated from WT and *Capn3*^{-/-} primary hepatocytes incubated at 45°C for 0, 15, 30, or 60 min. Western blot analysis was performed with indicated antibodies. This experiment was replicated at least twice.

Minor points:

- Fig. 1B, 5G: the y axis shows "fraction" but the legend says "percentage".

Authors' response: Thank you for pointing this out. We apologize for the oversight in the figure legend, where we have corrected the term "percentage" to "fraction".

- Fig. 1H: the alluvial plot is very hard to read and it is unclear what is the conclusion. Why is 16 h *Capn3* presented before 0 h? Why is there overall more state #4 at 16h than at 0h in *Capn3*? This is not consistent with the dynamics of single marks shown in Fig. 1E.

Authors' response: We acknowledge and apologize for any potential ambiguity in the original manuscript. The heatmaps depicted in the initial Fig. 1E were generated using peaks specific to each individual mark (H2AK119ub and H3K27me3) or transcription start sites (for H3K27ac). In contrast, the chromatin states were defined by segmenting the entire genome into 5 kb windows and integrating signals from all three marks. Each stream within the alluvial plot represents the same 5 kb genomic region across different conditions. We positioned the 16 h *Capn3*^{-/-} sample before the 0h time point, adjacent to the 16 h wild-type sample, to facilitate a direct comparison between the 16 h *Capn3*^{-/-} and 16 h wild-type conditions. The apparent increase of state #4 at 16 h *Capn3*^{-/-} compared with 0h does not contradict with the initial Fig. 1E; rather, it results from the integration of multiple marks across larger genomic windows, which can lead to shifts in chromatin state assignments that are not directly captured by single-mark peak dynamics. To better present the results, we have changed the color of state 3 from the similar shade as stat 4 to grey, in order to highlight the lines representing state 4 at 0h in wild-type.

These new data are now shown as Fig. 1E-G and described in the text of the paper.

Fig. 1E-G and legend:

Figure 1. Depletion of CAPN3 inhibits H2AK119ub remodeling and delays proliferation in hepatocytes after PHx

(E) Heatmap illustrating four chromatin states categorized by ChromHMM based on three histone modifications (H2AK119ub, H3K27me3 and H3K27ac). The intensity of the orange color indicates the enrichment of each ChIP-seq signal belonging to the given chromatin state.

(F) Box plot showing the expression levels of genes within each chromatin state in WT hepatocytes at 0 h post-PHx. The middle line denotes the median of Transcripts per million (TPM), the box represents the interquartile range, the whiskers denote 1.5 times the interquartile range from the upper or lower quartiles.

(G) Alluvial plots showing the global dynamics of chromatin states during liver regeneration (0 h vs 16 h post-PHx). Each line represents a 5 kb bin defined on the chromatin state categories

- Fig. 1J: *mistaken call in the text?*

Authors' response: Thank you for pointing this out. We apologize for our oversight and have corrected the erroneous reference to this panel.

- Fig. 2A: *I do not see a "drastic decrease". This should be repeated and quantified.*

Authors' response: Thank you for this comment. To address this concern, we have performed three additional independent biological replicates of these experiments. The results have been quantified and included in the revised manuscript as the new Fig.2A:

Fig. 2 and legend:

Figure 2. CAPN3 regulates H2AK119ub remodeling by proteolysis of PRC1 non-core components during liver regeneration.

(A) Western blot analysis of three independent treatments of Ctrl and Tet-HA-CAPN3 HepG2 cells with 1 μ g/mL doxycycline for 48 h (upper panel). Antibodies used are indicated. Lower panels show the quantification of relative expression levels of H2AK119ub (left) and H3K27me3 (right), normalized against Tubulin base on the western blot data. n = 3, **, p < 0.01.

- Fig. 2H: it seems that RYBP is decreased at 16h even in absence of CAPN3

Authors' response: We agree that RYBP is slightly decreased at 16 h even in absence of CAPN3. This observation aligns with the heat shock treatment results, where the dissociation of PRC1 from chromatin is delayed but still occurs in CAPN3 KO cells. These observations suggest the involvement of additional factors in this process. Consequently, we hypothesize that CAPN3 is crucial for the rapid response in this pathway.

We have incorporated this discussion into the "Discussion" section, as reprinted below:

Page 13, paragraph 1:

"However, as we overserved in both PHx and heat shock treatment, CAPN3 only delayed but not completely block the reduction of chromatin-bound PRC1, suggesting that additional pathways are involved in this process. Nonetheless, as we have consistently emphasized, CAPN3 is critically for the rapid response to stress signals."

- Fig. EV3 and EV5 seem to be missing

Authors' response: Thank you for pointing out this issue. We initially misunderstood the numbering guidelines and have now corrected and renumbered all figures accordingly.

- Fig. 5D-E: not called in the text

Authors' response: Thank you for pointing out this issue. We apologize for our

oversight and have now cited all the updated panels.

- A potential role for the H2AK119ub deubiquitinase PR-DUB should be discussed in the text.

Authors' response: We have incorporated this discussion into the "Discussion" section, as reprinted below:

Page 11, paragraph 2:

"Therefore, we proposed a model of the CAPN3-mediated PRC1 regulation system during H2AK119ub remodeling. The Polycomb repressive deubiquitinase (PR-DUB) complex plays a pivotal role in removing ubiquitin from H2AK119ub on nucleosomes and counteracts the ubiquitin E3 ligase function of PRC1 (Ge et al., 2023). On the other hand, the PRC1 complexes maintains H2AK119ub from deubiquitylation. After stress events such as PHx and heat shock, CAPN3 becomes activated and facilitates the dissociation of PRC1 complexes from chromatin through its proteolytic activity on non-core PRC1 components. Once a H2AK119ub modified region loses PRC1 occupancy, it becomes susceptible to rapidly deubiquitylation (Fig. 6G)."

- CAPN3 has low expression in the liver compared to tissue like muscle. Does this protein play a similar role in muscle growth or is this unique to hepatocytes and regeneration?

Authors' response: Thank you for raising this intriguing question. Although CAPN3 is predominantly expressed in cytoplasm of muscle tissue, and its function through titin and other substrates in muscle is well studied, we cannot rule out the possibility that the CAPN3-PRC1 pathway may also contribute to muscle growth or regeneration. We believe whether this pathway is unique to hepatocytes warrant further investigation and plan to explore in future studies.

- For IPs the % of input should be indicated in the figures.

Authors' response: Thank you for this suggestion. We have now included the '% of input' values in the relevant figure legends.

- Both the introduction and discussion mention "response to mitotic signals" being a key investigation in the paper, but only cell proliferation in general was discussed.

Authors' response: Thank you for pointing this out. We have now rephrased the relevant text to consistently focus on cell proliferation.

- The text should address the distinction between canonical and non-canonical PRC1 in regards to H2AK119ub deposition.

Authors' response: Thank you for your suggestion. We have now added the following paragraph to the "Introduction":

Page 3, paragraph 3:

"In mammals, PRC1 complexes are assembled around core components, either a

RING1A or RING1B dimerized with one of the six PcG finger (PCGF1-6) subunits. Canonical PRC1 complexes (cPRC1) containing CBXs and PHCs only form around either PCGF2 or 4, while noncanonical PRC1 complexes (ncPRC1) containing RYBP/YAF2 form around all the PCGFs (Piunti and Shilatifard, 2021). cPRC1 are recruited by H3K27me3 to compact chromatin and repress gene expression, whereas ncPRC1 are recruited independently of PRC2 and H3K27me3, and function as the most active E3 ubiquitin ligases (Zhao et al., 2020; Lopez et al., 2024).”

Dear Dr. Shi,

Thank you for submitting a revised version of your manuscript. Your study has now been seen by all original referees, who find that most of their previous concerns have been addressed and now recommend publication of the manuscript after some additional minor revisions. I find these remaining concerns constructive and reasonable and would therefore ask you to incorporate the requested changes into the final version of the manuscript. In addition, there remain only a few mainly editorial points that have to be addressed before I can extend formal acceptance of the manuscript:

- AUTHORS: name discrepancy: in ms: Hua Ruan vs. in eJP: Ruan Hua
- FUNDING INFO: missing in eJP: Hangzhou Chengxi Sci-tech innovation Corridor Management Committee; Shandong Provincial Laboratory Project (SYS202202) and Research Project of Jinan Microecological Biomedicine Shandong Laboratory (JNL-2025002B)
- KW (up to 5): in, 5, but should be placed below the Abstract
- Conflict Of Interest: title needs renaming to "DISCLOSURE AND COMPETING INTERESTS STATEMENT"
- As we are switching from a free-text author contribution statement towards a more formal statement based on Contributor Role Taxonomy (CRediT) terms, please remove the present Author Contribution section and instead specify each author's contribution(s) directly in the Author Information page of our submission system during upload of the final manuscript. See <https://casrai.org/credit/> for more information.
- Please complete the general info table in the authro checklist.
- DATASET EV LEGENDS: Dataset legend should be uploaded as a separate tab/sheet in the Excel file, and tab label should be updated to Dataset EV1 instead of Table S1
- The synopsis image is too large. It should be exactly 550 pixels wide and between 300-600 pixels high.
- Figure Legends (main + EV):
 1. Please note that the exact p values are not provided in the legends of figures 1A, B; 5C, E; 6B
 2. Please indicate the statistical test used for data analysis in the legends of figures 1A, B, J
 3. Please note that the box plots need to be defined in terms of minima, maxima, centre, bounds of box and whiskers, and percentile in the legend of figure 1F
 4. Please note that information related to n is missing in the legends of figures 5C, E; 6B
- Sections need to be named and the order should be corrected: Title page - Abstract - Keywords - Introduction - Results - Discussion - Methods - Data Availability - Acknowledgements - Disclosure and Competing Interests Statement - References - Figure Legends - Table(s) - Expanded View Figure Legends.

With best regards,
Cornelius Schneider

Cornelius Schneider, PhD
Editor | The EMBO Journal
c.schneider@embojournal.org

Please refer to our figure preparation guideline in order to ensure proper formatting and readability in print as well as on screen:

See also figure legend guidelines:

<https://www.embopress.org/page/journal/14602075/authorguide#figureformat>

Referee #1:

The authors have done excellent work. All my previous concerns were addressed, and the value of this study has been significantly enhanced.

I have only two minor issues that should be considered:

Regarding the added IP experiment in Fig 4E: The immunoblotting for KDM2B is not entirely convincing. Specifically, we could not see distinct signals for the long and short forms of KDM2B in the 'input' lane. However, I believe this data could be omitted/removed, as the strong signals for RING1B and PCGF3/5 are sufficient to robustly demonstrate that RYBPY70A is incorporated into the noncanonical PRC1.

Regarding Figure 2A: The sample descriptions in the lower panels of Figure 2A should be corrected to Control and HA-CAPN3 instead of 'WT' and 'Capn3-/-'."

Referee #2:

The authors have responded to each of our critiques on the content of the manuscript and have improved their paper of the course of this review process. I am overall satisfied with these revisions. Some minor concerns remain and are listed below.

- Fig. 1: it would be a useful addition if the authors would cross-analyze chromatin and RNA, specifically to show that genes with a delayed decrease in H2AK119ub are also those failing to be derepressed.
- The updated mass spec data are much improved. However, several PRC1 subunits (PCGF5, PCGF6, CBX6, and CBX7) that were present previously are no longer shown. It would be important to at least show results for CBX6, which was identified as a CAPN3 substrate.
- Previous point 8: why is the autolysis of CAPN3 no evident in all western blots? For example I do not see it in Fig. 3B. Please explain. Along similar lines, in Fig. EV2D WT and C129S appear similarly abundant, whereas in Fig. EV2F, the WT protein is almost absent. Can the authors provide an explanation or speculate about the source of variability?
- Additionally, it remains unclear why the C129S mutation causes such a large apparent size shift by SDS-PAGE, given that the only difference between the proteins is a cysteine-to-serine substitution.
- Previous point 10, Fig. 6D: it would be good to include fractionation controls, i.e. protein markers for each compartment analyzed.

Response to Reviewer #1:

We thank reviewer 1 for the positive assessment and insightful suggestions regarding our revision. We have now clarified each of the new issues raised, and a point-by-point response is provided below:

Reviewer 1 comment: *The authors have done excellent work. All my previous concerns were addressed, and the value of this study has been significantly enhanced. I have only two **minor issues** that should be considered:*

1. Regarding the added IP experiment in Fig 4E: The immunoblotting for KDM2B is not entirely convincing. Specifically, we could not see distinct signals for the long and short forms of KDM2B in the 'input' lane. However, I believe this data could be omitted/removed, as the strong signals for RING1B and PCGF3/5 are sufficient to robustly demonstrate that RYBPY70A is incorporated into the noncanonical PRC1.

Authors' response: We thank the reviewer for this suggestion. We agree that the strong signals for RING1A/B and PCGF3/5 are sufficient to robustly demonstrate that RYBP^{Y70A} is incorporated into the noncanonical PRC1. Therefore, we have removed the anomalous data for KDM2B. The revised Fig. 4E is provided below:

Fig. 4E and legend:

Figure 4. Tyr⁷⁰ of RYBP was the key amino acid for its proteolysis by CAPN3

(E) Western blot analysis on anti-Flag immunoprecipitated from nuclear extracts of 293T cells transfected with 500 ng *Flag-RYBP-pCS2+*, 500ng *Flag-RYBP^{Y70A}-pCS2+* or 500ng *pCS2+*. Cell lysates were harvested after 48 h. 'Input' contains ~10% of the input cell lysate used for IP. Antibodies used are indicated.

2. Regarding Figure 2A: The sample descriptions in the lower panels of Figure 2A should be corrected to Control and HA-CAPN3 instead of 'WT' and 'Capn3-/-'.

Authors' response: We thank the reviewer for pointing out this issue. We apologize for the oversight and have corrected the sample descriptions accordingly.

Fig. 2A and legend:

Figure 2. CAPN3 regulates H2AK119ub remodeling by proteolysis of PRC1 non-core components during liver regeneration.

(A) Western blot analysis of three independent treatments of Ctrl and Tet-HA-CAPN3 HepG2 cells with 1 μ g/mL doxycycline for 48 h (upper panel). Antibodies used are indicated. Lower panels show the quantification of relative expression levels of H2AK119ub (left) and H3K27me3 (right), normalized against Tubulin base on the western blot data. Data represent mean \pm s.d. (n=3). The statistical analysis was done using two-tailed Student's *t*-test. A *p* value < 0.05 was considered statistically significant, and exact *p* values (*p*) are indicated in the figure. Source data are provided in the source data file.

Response to Reviewer #2:

We thank reviewer 2 for the positive assessment and insightful suggestions regarding our revision. We have now clarified each of the new issues raised, and a point-by-point response is provided below:

Reviewer 2 comment: *The authors have responded to each of our critiques on the content of the manuscript and have improved their paper of the course of this review process. I am overall satisfied with these revisions. Some **minor concerns** remain and are listed below.*

1. Fig. 1: *it would be a useful addition if the authors would cross-analyze chromatin and RNA, specifically to show that genes with a delayed decrease in H2AK119ub are also those failing to be derepressed.*

Authors' response: We thank the reviewer for this insightful suggestion and apologize for any lack of clarity in our previous response and the original manuscript regarding this point. In our analysis, we defined gene clusters #41_44, #42_44 and #43_44 as those exhibiting a delayed decrease in H2AK119ub levels in *Capn3*^{-/-} hepatocytes at 16 h post-PHx. Accordingly, Fig. 1I and Fig. 1J each present a subset of genes within these clusters that fail to undergo transcriptional derepression at 16 h and 24 h post-PHx, respectively. We have included further clarifications in the revised manuscript.

The revised sentences are reprinted below:

Figure 1I legend:

“(I) Venn diagrams showing the extent of overlap for genes near #41_44, #42_44 and #43_44 loci that display a delayed reduction in H2AK119ub, genes upregulated in WT hepatocytes at 16 h post-PHx vs. 0 h, and genes downregulated in *Capn3*^{-/-} hepatocytes at 16 h post-PHx relative to WT. The intersection of these three gene sets is highlighted by the red dotted-line frame.”

Figure EV2H legend:

“(H) Covariance clustering of genes near #41_44, #42_44 and #43_44 loci that display a delayed reduction in H2AK119ub (TPM>0.1 in ≥ 1 condition, n=1232) across 16, 24, and 36 h PHx in WT and *Capn3*^{-/-} hepatocytes into 16 clusters. Gene counts for each cluster are shown at the bottom right of each corresponding panel.”

2. The updated mass spec data are much improved. However, several PRC1 subunits (PCGF5, PCGF6, CBX6, and CBX7) that were present previously are no longer shown. It would be important to at least show results for for CBX6, which was identified as a CAPN3 substrate.

Authors' response: We thank the reviewer for raising this issue. Peptides corresponding to PCGF5, PCGF6, CBX6, and CBX7 were not detectable in this mass spectrometry analysis. This is likely due to differences in detection depth and sensitivity between different runs. To ensure transparency, we have provided the complete list of proteins identified in this experiment as a source data file (Fig.2B_Ms.xlsx).

3. Previous point 8: why is the autolysis of CAPN3 no evident in all western blots? For example I do not see it in Fig. 3B. Please explain. Along similar lines, in Fig. EV2D WT and C129S appear similarly abundant, whereas in Fig. EV2F, the WT protein is almost absent. Can the authors provide an explanation or speculate about the source of variability?

Authors' response: We thank the reviewer for this important question and the opportunity to clarify. CAPN3 undergoes substantial autolysis intracellularly. Therefore, for the *in vitro* CAPN3 proteolysis assay, cells overexpressing CAPN3 were lysed at 20 h post-transfection to obtain workable protein levels, rather than at 48 h as for substrate proteins (see Materials and Methods). Further autolysis can sometimes be observed during the assay (e.g. source data for Fig. 2E of the *in vitro* CAPN3 proteolysis assay for RING1B and some uncropped data bellow), but it is not always detectable.

We also recognized that the variation in CAPN3 levels between assay batches could affect comparisons across different PRC1 subunits. To ensure reliability, each batch included 1-4 candidate substrates, along with a CAPN3-resistant negative control and a CAPN3-susceptible positive control. Each candidate substrate was tested in at least

two independent experiments.

Please find the relevant uncropped data below:

Figure for reviewer: Western blot analysis of *in vitro* CAPN3 proteolysis assays for PCGF2 and PCGF6, with uncropped CAPN3 membranes shown. Further CAPN3 autolysis products are indicated by red asterisks.

4. Additionally, it remains unclear why the C129S mutation causes such a large apparent size shift by SDS-PAGE, given that the only difference between the proteins is a cysteine-to-serine substitution.

Authors' response: We thank the reviewer for this insightful question. The C129 residue is critical to the catalytic core of CAPN3, and the C129S mutation abolishes both its proteolytic activity and autolysis. As mentioned above, the wild-type CAPN3 detected by western blot primarily represents a partially autolyzed form, which migrates faster than the full-length protein. In contrast, the catalytically inactive C129S mutant remains stable and intact due to the absence of autolysis. This difference likely accounts for the apparent size shift observed by SDS-PAGE. To confirm this, we performed western blot analysis of the *in vitro* CAPN3 proteolysis assay for two candidate substrates. The anti-HA antibody detects intact CAPN3 bands, which show similar sizes for both wild-type and C129S mutant, although the wild-type CAPN3 bands are much weaker. After stripping the membrane, the anti-CAPN3 antibody (the same one used in the manuscript) was applied, revealing predominantly the partially autolyzed form, which migrates faster than the full-length protein, consistent with the assays in the manuscript.

These new data are now shown below:

Figure for reviewer: Western blot analysis of the *in vitro* CAPN3 proteolysis assay for two candidate substrates. The antibodies used for detection are indicated. The anti-CAPN3 antibody applied following stripping of the anti-HA antibody.

5. Previous point 10, Fig. 6D: it would be good to include fractionation controls, i.e. protein markers for each compartment analyzed.

Authors' response: We thank the reviewer for this valuable suggestion. We have now included fractionation controls: α -Tubulin was detected only in total cell extracts, whereas GAPDH was present in total and nuclear extracts but absent from chromatin fractions. The updated Fig. 6D is shown below:

Fig. 6D and legend:

Figure 6. Proteolysis of PRC1 non-core components by CAPN3 disrupts the maintenance of genomic H2AK119ub by PRC1 occupation

(D) Total cell extracts, nuclear extracts and Chromatin fractions were isolated from WT and *Capn3*^{-/-} primary hepatocytes incubated at 45°C for 0, 15, 30, or 60 min. Western blot analysis was performed with indicated antibodies. This experiment was replicated at least twice.

Dear Dr. Shi,

I am pleased to inform you that your manuscript has been accepted for publication in the EMBO Journal.

You may qualify for financial assistance for your publication charges - either via a Springer Nature fully open access agreement or an EMBO initiative. Check your eligibility: <https://link.springer.com/journal/44318/how-to-publish-with-us>

Yours sincerely,

Cornelius Schneider, PhD
Editor
The EMBO Journal
c.schneider@embojournal.org

Please note that it is The EMBO Journal policy for the transcript of the editorial process (containing referee reports and your response letters) to be published as an online supplement to each paper. If you should prefer removal of any referee-only figures included in the point-by-point response(s), e.g. because they may still be used for future publication or because they have been reproduced from published work by others, please do let us know immediately via response email.

More information is available here: <https://link.springer.com/partners/embo-press/editorial-policies#Peer%20review>